# Binding blockade between TLN1 and integrin β1 represses triple-negative breast cancer

Yixiao Zhang[1,2†], Lisha Sun[1,2,3*†], Haonan Li[4†], Liping Ai[2], Qingtian Ma[1,2], Xinbo Qiao[1,2], Jie Yang[2], Hao Zhang[2], Xunyan Ou[2], Yining Wang[2], Guanglei Chen[1,2], Jinqi Xue[1,2], Xudong Zhu[1,2], Yu Zhao[5], Yongliang Yang[1,4*], Caigang Liu[1,2,3*]

[1]Department of Oncology, Shengjing Hospital of China Medical University, Shenyang, China; [2]Cancer Stem Cell and Translational Medicine Laboratory, Shengjing Hospital of China Medical University, Shenyang, China; [3]Innovative Cancer Drug Research and Development Engineering Center of Liaoning Province, Shenyang, China; [4]School of Bioengineering, Dalian University of Technology, Dalian, China; [5]Department of Biochemistry and Molecular Biology, Mayo Clinic, Rochester, United States

*For correspondence:
sunlisha1224@126.com (LS);
everbright99@foxmail.com (YY);
angel-s205@163.com (CL)

†These authors contributed equally to this work

## Abstract

**Background:** Integrin family are known as key gears in focal adhesion for triple-negative breast cancer (TNBC) metastasis. However, the integrin independent factor TLN1 remains vague in TNBC.
**Methods:** Bioinformatics analysis was performed based on TCGA database and Shengjing Hospital cohort. Western blot and RT-PCR were used to detect the expression of TLN1 and integrin pathway in cells. A small-molecule C67399 was screened for blocking TLN1 and integrin β1 through a novel computational screening approach by targeting the protein-protein binding interface. Drug pharmacodynamics were determined through xenograft assay.
**Results:** Upregulation of TLN1 in TNBC samples correlates with metastasis and worse prognosis. Silencing *TLN1* in TNBC cells significantly attenuated the migration of tumour cells through interfering the dynamic formation of focal adhesion with integrin β1, thus regulating FAK-AKT signal pathway and epithelial-mesenchymal transformation. Targeting the binding between TLN1 and integrin β1 by C67399 could repress metastasis of TNBC.
**Conclusions:** TLN1 overexpression contributes to TNBC metastasis and C67399 targeting TLN1 may hold promise for TNBC treatment.

**Funding:** This study was supported by grants from the National Natural Science Foundation of China (No. 81872159, 81902607, 81874301), Liaoning Colleges Innovative Talent Support Program (Name: Cancer Stem Cell Origin and Biological Behaviour), Outstanding Scientific Fund of Shengjing Hospital (201803), and Outstanding Young Scholars of Liaoning Province (2019-YQ-10).

## Editor's evaluation

The paper is of interest to preclinical and translational scientists in the field of breast cancer. It details the identification, characterization and selection of a novel drug candidate, based on the biology of a select target gene, with the premise to rationalize a new therapy for triple-negative breast cancer. The data presented support the proposed hypotheses, and the conclusions are well supported by the results.

## Introduction

Triple-negative breast cancer (TNBC) is characterized by absence of hormone receptor and human epidermal growth factor receptor 2 (*HER2*) amplification, which constitutes 15–20% of breast cancers with the highest mortality among all breast cancer subtypes (*Arroyo-Crespo et al., 2019*; *Kim et al., 2018*). In contrast to hormone receptor positive breast cancer (luminal subtype), targeted therapy towards TNBC still lacks and conventional chemotherapy remains the standard care for TNBC patients (*Wahba and El-Hadaad, 2015*). TNBC targeted therapies have been actively developed and approved by FDA, including PARP inhibitors for BRCA-1 mutated breast cancer, perbrolizumab-based immunotherapy, and Sacituzumab for Trop-2 (*Vagia et al., 2020*), but new therapeutic strategies are urgently needed to tackle this aggressive tumour for TNBC patients.

Unfortunately, the pathogenesis underlying the aggressive behaviour of TNBC is poorly understood. Epithelial-mesenchymal transformation (EMT) induces epithelial cells to acquire mesenchymal phenotypes, resulting in tumour progression and metastasis (*Sikandar et al., 2017*). Previous studies have proven that progression of EMT is critical for aggression of TNBC (*Xu et al., 2012*; *Zhao et al., 2014*). Hence, understanding potential molecular mechanisms will be of great significance to uncover new therapeutic targets for the management of patients with TNBC.

It is well known that invasive cancer cells adhere to extracellular molecules and disrupt the extracellular matrix (ECM) to initiate the metastatic process (*Ishaque et al., 2018*). TLNs are cytoplasmic adapter proteins essential for integrin-mediated cell adhesion to the ECM. Moreover, TLNs are responsible for the activation of integrins via linking integrins to cytoskeletal actin and coordinating recruitment of microtubules to adhesion sites (*Gough and Goult, 2018*). Loss of TLNs can impair EMT and the acquisition of cell motility (*Thapa et al., 2017*). There are two isoforms of TLNs: TLN1 and TLN2 (*Gough and Goult, 2018*). TLN1 is expressed in almost all tissues, while TLN2 is usually expressed mainly in the heart, brain, testis, and muscle (*Debrand et al., 2009*; *Manso et al., 2013*). Most of attention on TLNs has focused on TLN1 due to its essential role in mediating cell adhesion. TLN1 is located in focal adhesion (FA), which regulates integrin signalling and promotes metastasis in different cancers (*Desiniotis and Kyprianou, 2011*; *Hoshino et al., 2015*; *Klapholz and Brown, 2017*; *Seguin et al., 2015*), including prostate cancer (*Jin et al., 2015*), colon cancer (*Bostanci et al., 2014*), and oral squamous cell carcinoma (*Lai et al., 2011*). However, the biological effect of TLN1 deletion on the malignancy of TNBC as well as the underlying molecular mechanism remains vague.

Recent studies have reported that the auto-inhibition of TLN1 controls cell-ECM (CEM) adhesion, migration, and wound healing in vivo (*Haage et al., 2018*), and disrupting this auto-inhibition leads to more mature and stable FAs (*Haage et al., 2018*). Other studies have also shown that the rapid transition between the active and inactive conformations of TLN1 regulates FA turnover, a critical process for cell adhesion and signal transduction (*Dedden et al., 2019*). Moreover, a previous study found higher *TLN1* transcripts in TNBC than in other clinical breast cancer subtypes (*Singel et al., 2013*). Therefore, disruption of TLN1 might be a potential strategy for cancer therapy.

Herein, we first explored the role of TLN1 in the pathogenesis of TNBC. We investigated the expression of TLN1 in TNBC cells and studied the effects of silencing *TLN1* on TNBC cell proliferation, adhesion, and migration in vitro and in vivo. In addition, for the first time, we identified a small-molecule compound through a novel computational screening approach by targeting protein-protein binding interface (CSTPPI) to block the interaction between TLN1 and integrin β1. Moreover, we assessed the interference of small-molecule inhibitors with FAs formation and migration in vitro, as well as the inhibition of tumour size and lung metastasis in vivo. Our findings suggest TLN1/integrin β1 binding as a potential therapeutic target for TNBC.

## Materials and methods

### Key resources table

| Reagent type (species) or resource | Designation | Source or reference | Identifiers | Additional information |
|---|---|---|---|---|
| Cell line (*Homo sapiens*) | MDA-MB-231 | ATCC | ATCC Cat# HTB-26, RRID:CVCL_0062 | |
| Cell line (*Homo sapiens*) | MCF-7 | ATCC | ATCC Cat# CRL-3435, RRID:CVCL_A4AQ | |

*Continued on next page*

*Continued*

| Reagent type (species) or resource | Designation | Source or reference | Identifiers | Additional information |
|---|---|---|---|---|
| Cell line (*Homo sapiens*) | BT-549 | ATCC | ATCC Cat# HTB-122, RRID:CVCL_1092 | |
| Cell line (*Homo sapiens*) | SK-BR-3 | ATCC | ATCC Cat# HTB-30, RRID:CVCL_0033 | |
| Transfected construct (human) | TLN1 shRNA #1 | Sangon Biotech | | Transfected construct (human) |
| Antibody | Anti-Vinculin (Rabbit polyclonal) | Proteintech, Rosemont, IL | Cat# 26520–1-AP, RRID:AB_2868558 | WB (1:1000), IF(1:200) |
| Antibody | Anti-E-cadherin (Rabbit monoclonal) | Cell Signaling Technology, Danvers, MA | Cat# 3195, RRID:AB_2291471 | WB (1:1000) |
| Antibody | Anti-N-cadherin (Rabbit polyclonal) | Cell Signaling Technology, Danvers, MA | Cat# 4061, RRID:AB_10694647 | WB (1:1000) |
| Antibody | Anti-Tubulin α (Rabbit monoclonal) | Cell Signaling Technology, Danvers, MA | Cat# 2125, RRID:AB_2619646 | WB (1:5000) |
| Antibody | Anti-CK18 (mouse monoclonal) | Cell Signaling Technology, Danvers, MA | Cat# 4546, RRID:AB_2134843 | WB (1:1000) |
| Antibody | Anti-FAK (Rabbit polyclonal) | Cell Signaling Technology, Danvers, MA | Cat# 3285, RRID:AB_2269034 | WB (1:1000) |
| Antibody | Anti-GAPDH (Rabbit polyclonal) | Proteintech, Rosemont, IL | Cat# 10494–1-AP, RRID:AB_2263076 | WB (1:5000) |
| Antibody | Anti-AKT1 (Rabbit polyclonal) | Proteintech, Rosemont, IL | Cat# 10176–2-AP, RRID:AB_2224574 | WB (1:1000) |
| Antibody | Anti-integrin β3 (Rabbit polyclonal) | Proteintech, Rosemont, IL | Cat# 18309–1-AP, RRID:AB_2128759 | WB (1:1000) |
| Antibody | Anti-p-AKT1 (sc-81433) (mouse monoclonal) | Santa Cruz Biotechnology, Dallas, TX | Cat# sc-81433, RRID:AB_1125472 | WB (1:1000) |
| Antibody | Anti-p-FAK (Tyr397) (Rabbit polyclonal) | Immunoway, Plano, TX | Cat# YP0739, RRID:AB_2904589 | WB (1:1000) |
| Antibody | Anti-integrin β1 (Rabbit polyclonal) | Sino Biological, China | Cat# 100562-T46, RRID:AB_2895614 | WB (1:1000), IF (1:200) |
| Antibody | Anti-TLN1 (Rabbit monoclonal) | Cell Signaling Technology, Danvers, MA | Cat# 4021, RRID:AB_2204018 | WB (1:1000), IF (1:200), IHC(1:500) |
| Antibody | Goat Anti-Rabbit IgG H&L Alexa Fluor 488 (Goat polyclonal) | Abcam, Waltham, MA | Cat# ab150077, RRID:AB_2630356 | IF (1:400) |
| Antibody | Goat Anti-Mouse IgG H&L Alexa Fluor 647 (Goat polyclonal) | Abcam, Waltham, MA | Cat# ab150115, RRID:AB_2687948 | IF (1:400) |
| Commercial assay or kit | CCK-8 kit | Dojindo, Japan | CK04 | |
| Commercial assay or kit | Cell adhesion detection kit | Best Bio, Nanjing, China | BB-48120 | |
| Commercial assay or kit | Annexin-V and propidium iodide (PI) kit | BD, San Diego, CA | 559763 | |
| Commercial assay or kit | EdU kit | Ribobio, China | C10327 | |
| Chemical compound, drug | C67399 | Chemdiv Compond, San Diego, CA | 4903–2135 | C38H48O5, 584.79 Da |
| Software, algorithm | SPSS | SPSS | RRID:SCR_002865 | |
| Software, algorithm | GraphPad Prism | GraphPad Prism | RRID:SCR_002798 | |
| Software, algorithm | ImageJ | ImageJ | RRID:SCR_003070 | |
| Software, algorithm | PyMOL | PyMOL | RRID:SCR_000305 | |
| Software, algorithm | FIPSDock | FIPSDock | PMID:22961860 | |
| Other | DAPI stain | UE, China | D4054 | (5 μg/ml) |
| Other | Rhodamine-labelled phalloidin stain (TRITC Phalloidin) | Solarbio, China | Cat# CA1610, RRID:AB_2904593 | (100 nM) |

| Reagent type (species) or resource | Designation | Source or reference | Identifiers | Additional information |
|---|---|---|---|---|
| Other | Protein A/G agarose beads | Santa Cruz Biotechnology, Dallas, TX | Cat# sc-2003, RRID:AB_10201400 | (50 µl/sample) |

## Patients

A total of 171 patients with TNBC were recruited at Shengjing Hospital of China Medical University. Patients with breast cancer were diagnosed based on histological examinations. When the patients underwent radical mastectomy, breast tumour and para-tumour tissues and at least 10 lymph nodes were collected and pathologically examined. The inclusion criteria were as follows: patients had complete clinical data, did not receive chemotherapy or radiotherapy before surgery, no other types of malignant tumours, no severe organ dysfunction, and no bilateral breast cancer. Tissues were fixed in 10% formalin and paraffin-embedded for histological examination and immunohistochemistry. Some tissues were snap-frozen in liquid nitrogen for protein expression analyses. Written informed consent was obtained from all the patients, and this study was approved by the institutional research ethics committee of Shengjing Hospital of China Medical University (Project identification code: 2018PS304K, dated on 03/05/2018).

## Immunohistochemistry

The expression of TLN1 in breast tumour, para-tumour, and metastatic tissues was examined by immunohistochemistry. Briefly, the breast tumour and para-tumour tissue sections (4 µm) were fixed, paraffin-embedded, dewaxed, and rehydrated as mentioned on Histology. The tissue sections were repaired with citrate antigen repair buffer (pH = 6.0) and heating in a microwave for 10 min. After incubating in 3% hydrogen peroxide solution, the sections were blocked with 5% bovine serum albumin (BSA) at 37°C for 1 hr, followed by incubation overnight with primary antibodies against TLN1 (4021; Cell Signaling Technology, Danvers, MA) with final dilution of 1:100. After washing with PBS, the sections were incubated with horseradish peroxidase (HRP)-conjugated goat-anti-IgG at room temperature for 30 min, the binding of which was visualized with 3,3′-diaminobenzidine, and counterstained with haematoxylin before they were mounted and imaged under a light microscope (Olympus, Japan). The intensity and frequency of positively stained cells were evaluated in a blinded manner (*Liu et al., 2018*).

The immunohistochemical signals were assessed independently by two pathologists in a blinded manner as before (*Gu et al., 2018*). Briefly, a total of 200 tumour cells in two 400× microscope fields of each specimen were evaluated for the percentages of positively stained cells. The staining intensity was scored as: 0 (colourless), 1 (light yellow), 2 (brownish yellow), or 3 (brown). The percentages of stained cells were scored as: 0 (<5% positive cells), 1 (5–25% positive cells), 2 (26–50% positive cells), 3 (51–75% positive cells), or 4 (>75% positive cells). A final score (intensity × percentage) was calculated and classified as: negative (score of 0), weakly positive (score of 1–4), moderately positive (score of 5–8), or strongly positive expression (score of 9–12). All specimens were stratified as low expression of TLN1 (scores of 0–4) or high expression of TLN1 (scores of 5–12) (*Gu et al., 2018*).

## TCGA analysis

UALCAN, an interactive web-portal to perform to in-depth analyses of The Cancer Genome Atlas (TCGA) gene expression data (http://ualcan.path.uab.edu) (*Chandrashekar et al., 2017*) was used to analyse *TLN2* mRNA expression in breast cancer and normal breast tissue, as well as survival between different breast cancer subtypes. The correlation of TLN1 with key regulators in EMT pathway was performed in GEPIA (gene expression profiling interactive analysis, http://gepia.cancer-pku.cn/). Correlation between TLN1 and the integrin family in different types of the TCGA-BRCA dataset was obtained in the TIMER database.

## Cell culture

MDA-MB-231, MCF-7, BT-549, and SK-BR-3 human breast cancer cells were obtained from the American Type Culture Collection (ATCC, Manassas, VA). MDA-MB-231 cells were cultured in Leibovitz's L15 medium and BT-549 cells were cultured in RPMI 1640 medium, MCF-7 cells were cultured in DMEM, and SK-BR-3 cells were cultured in McCoy's 5A medium (Thermo Fisher, Waltham, MA), supplemented

with 10% foetal bovine serum (Cellmax, China), penicillin (100 units/ml), and streptomycin (100 µg/ml). All cells were incubated at 37°C in a humidified atmosphere of 5% $CO_2$. None of the cell lines were contaminated with mycoplasma.

## Transduction

To establish MDA-MB-231/NC and MDA-MB-231/shRNA cells with stable knockdown of TLN1 expression, MDA-MB-231 cells were transduced with lentivirus (at a multiplicity of infection of 10) containing either control shRNA (NC) or TLN1-specific shRNA (Sangon Biotech, Shanghai, China) and cultured in the presence of 5 µg/ml puromycin (Thermo Fisher, Waltham, MA) for 4 days for selection. The knockdown efficiency was determined by western blotting (described in a subsequent section). The TLN1-specific shRNA sequence was 5'-GCAGTGAAAGATGTAGCCAAA-3'.

## EdU assay

The proliferation of cells was determined using an EdU kit (C10327, Ribobio, Guangzhou, China) according to the manufacturer's instructions. Briefly, MDA-MB-231/NC and shTLN1 cells ($4 \times 10^3$ cells/well) were cultured in serum-free medium for 12 hr in triplicate in 96-well plates before the medium was changed to complete medium. After culture for 24, 48, or 72 hr, the cells were labelled with 50 µM EdU during the last 2 hr of culture. Then, the cells were washed with PBS, fixed in 4% paraformaldehyde, permeabilized with 0.5% Triton X-100 at room temperature for 10 min, and treated with additive solution buffer for 30 min in the dark. After washing again, the cells were stained with the nuclear dye Hoechst 33342 at 1:2000 and imaged under a fluorescence microscope. The percentages of EdU+ cells among all Hoechst 33342+ cells were determined in a blinded manner and calculated using ImageJ software (NIH, Bethesda, MD).

## Adhesion assay

The effect of silencing *TLN1* on the adhesion of MDA-MB-231 cells was determined using a cell adhesion detection kit (BB-48120, Best Bio, Nanjing, China) according to the manufacturer's instructions. Briefly, MDA-MB-231/NC and shTLN1 cells ($5 \times 10^4$ cells/well) were seeded in triplicate in fibronectin/laminin-precoated 96-well plates and cultured at room temperature for 1 hr. In some experiments, the cells were pre-treated with vehicle (DMSO) or 2.0 µM C67399 (AO-774/41465499, Specs, San Francisco, CA) for 24 hr prior to the adhesion assay. After the cells were extensively washed with PBS, they were cultured in the presence of 20 µl of CCK-8 solution for 2 hr, and the absorbance was measured at 450 nm using a microplate reader (Titertek Multiskan PLUS, MK II, Labsystems, Waltham, MA,USA). The computer program PRISM and ImageJ were used to create graphs, process images, and perform statistical analysis.

## Immunofluorescence

MDA-MB-231 cells ($4 \times 10^3$ cells/well) were serum-starved overnight and then cultured in complete medium in 24-well plates for 18 hr. The cells were stained with anti-integrin β1 (00562-T46, SinoBio, China) and anti-TLN1 (26520–1-AP, Proteintech, Rosemont, IL) at final dilution of 1:200 overnight at 4°C, followed by Alexa Fluor 647 (ab150115, Abcam) and Alexa Fluor 488 (ab150077, Abcam) staining. Or stained with Alexa Fluor 488 (ab150077, Abcam)-labelled anti-vinculin (26520–1-AP, Proteintech, Rosemont, IL) with final dilution of 1:200 overnight at 4°C and washed with PBS. And stained with rhodamine-labelled phalloidin (CA1610, Solarbio, China) with final dilution of 1:200 for 1 hr at room temperature, followed by nuclear staining with DAPI (5 µg/ml, UE, D4054, China). The cells were imaged under a confocal microscope (Zeiss, LSM800). The cell length and numbers and sizes of FAs were measured for 20 randomly selected cells from each group in a blinded manner.

## Flow cytometry

The effect of TLN1 silencing on spontaneous apoptosis in TNBC cells was measured by flow cytometry using an Annexin-V and propidium iodide (PI) kit (559763, BD, San Diego, CA), following the protocol provided. Briefly, the different groups of cells ($2 \times 10^5$/tube) were stained in duplicate with 5 µl of Annexin V-FITC and PI in the dark for 15 min. The cells were then analysed by flow cytometry (BD FACSverse, Piscataway, NJ ).

## Transmission electron microscopy

The ultrastructure of MDA-MB-231/NC and MDA-MB-231/shTLN1 cells was examined by transmission electron microscopy. Briefly, the different groups of cells were harvested, fixed in 2.5% glutaraldehyde, and embedded. Ultra-thin sections (70 nm) were prepared using a microtome and mounted on a copper grid. The sections were stained with 4% aqueous uranyl acetate (10 min) and then Reynolds lead citrate (2 min). The cells were photoimaged under a transmission electron microscope (JEM-2000EX, JEOL, Sagamihara, Japan).

## Western blotting

The protein levels of molecules of interest relative to those of GAPDH (the loading control) in different groups of cells were determined by western blotting (*Liu et al., 2018*). Briefly, $5 \times 10^6$ of MDA-MB-231/NC and shTLN1 cells were lysed in RIPA buffer for 30 min at 4°C. After determining protein concentrations using a BCA kit (Thermo Fisher, Waltham, MA), the cell lysates (40 µg/lane) were separated by sodium dodecyl sulfate polyacrylamide gel electrophoresis on 8–12% gels and transferred to polyvinylidene difluoride membranes (Millipore, Sigma, Burlington, MA). The membranes were blocked with 5% BSA in TBST for 1 hr at room temperature and incubated with primary antibodies at 4°C overnight. The primary antibodies were anti-TLN1 (4021), anti-vimentin (5741), anti-E-cadherin (3195), anti-N-cadherin (4061), anti-tubulin α (2125), anti-CK18 (4546), anti-FAK (3285) purchased from Cell Signaling Technology (Danvers, MA); anti-GAPDH (10494–1-AP), anti-AKT1 (10176–2-AP), anti-integrin β3 (18309–1-AP) obtained from Proteintech (Rosemount, IL); anti-p-AKT1 (Santa Cruz Biotechnology, sc-81433, Dallas, TX); anti-p-FAK (Tyr397, Immunoway, YP0739, Plano, TX); and anti-integrin β1 (Sino Biological, 100562-T46, China). The membranes were then treated with HRP-conjugated secondary antibodies (1:10,000 dilution; Jackson ImmunoResearch Laboratories, West Grove, PA), and the proteins were visualized using enhanced chemiluminescence reagents (Thermo Fisher, Waltham, MA). The levels of individual target proteins relative to those of GAPDH or tubulin α were determined based on densitometric analysis using ImageJ software.

## Immunoprecipitation

MDA-MB-231 cells ($1 \times 10^7$) were lysed in cold RIPA lysis buffer containing proteinase inhibitors. Lysis is followed by a washing step to remove cell debris and the lysis buffer. The cell lysates (50 µg/tube) were then incubated with anti-TLN1, anti-integrin β1, anti-integrin β3, or control isotype IgG (2 µg) with gentle agitation at 4°C overnight. Subsequently, immunocomplexes were precipitated with 50 µl of protein A/G agarose beads (sc-2003, Santa Cruz Biotechnology, Dallas, TX) at 4°C for 4 hr. After centrifugation at 700 *g* and at 4°C for 5 min, the microbeads were washed with lysis, and the bound proteins were disentangled from antigen-antibody-beads complex by heating at 100°C for 5 min. The proteins were analysed by western blotting.

## Structural simulation and targeted molecular screening of TLN1

The TLN1 protein contains F0, F1, F2, and F3 domains in the atypical FERM (band 4.1, ezrin, radixin, and moesin) structure (PDB ID: 3IVF). Given that the S1 and S2 chains of the phosphotyrosine-binding (PTB) F3 domain are crucial for integrin binding and activation (*Bouaouina et al., 2008*; *Tadokoro et al., 2003*; *Wegener et al., 2007*), the F3 domain of TLN1 was targeted. The flexible loop domain was labelled in cartoon mode and surface mode using PyMOL software. The potential ligands interacting with the PTB F3 domain of TLN1 were virtually screened using the FIPSDock tool (*Liu et al., 2013*). The optimal scoring of each molecule and the corresponding docking conformation and pocket mode of action, where the scoring was related to the negative logarithm (−logKd) of the ligand-receptor dissociation equilibrium constant, were assessed. The binding mode of the complexes identified from docking was used as a baseline structure for 5 ns of molecular dynamic (MD) simulations using the academic free package Gromacs (*Wang et al., 2018*).

## Transwell assays

The effect of silencing *TLN1* on the migration and invasion of MDA-MB-231 cells was determined with transwell migration and invasion assays (*Gu et al., 2019*). Briefly, MDA-MB-231/NC and shTLN1 cells ($10^5$ cells/well) were seeded in duplicate in serum-free medium in the top chamber of transwell plates (8 µm pore size, Corning) and cultured for 24 hr. The bottom chambers were filled with complete

medium. For the invasion assays, the membranes were coated with Matrigel in the upper chambers. The cells on the membranes of the top chamber were fixed by methyl alcohol for 20 min and stained by 0.1% crystal violet for 20 min at room temperature. Initial cell status was visualized immediately using a light microscope (Olympus, Tokyo, Japan). Cells in five fields per membrane were randomly selected and counted in a blinded manner.

## Mass spectrometry

The effect of silencing *TLN1* on differentially expressed proteins (DEPs) in MDA-MB-231 cells was determined by tandem mass tag (TMT, Thermo, Pierce Technology, Waltham, MA) and LC-MS/MS. Briefly, proteins from MDA-MB-231/NC and shTLN1 cells were extracted. Three biological replicates were generated for each group. After labelled with TMT reagents for 2 hr at room temperature and then pooled and desalted, the peptide samples were subjected to LC-MS/MS using a Q Exactive system (Thermo Fisher, Waltham, MA) with a C18 column. The protein profiles with $p < 0.05$ and the difference multiple more than 1.2 times or less than 0.83 times were selected as DEPs. Then DEPs were analysed by GO analysis using WebGestaltR (http://www.webgestalt.org/option.php).

## Xenograft tumours in mice

Xenografted TNBC tumours were established as previously described (*Liu et al., 2018*). Briefly, $10^6$ MDA-MB-231/NC or shTLN1 cells were injected into the mammary fat pads of NOD/SCID mice (n = 8 per group). The growth of the implanted tumour diameter was monitored longitudinally for up to 21 days post-inoculation, at which point the mice were sacrificed. The tumours were dissected, measured, and weighed.

For lung metastases, $10^6$ MDA-MB-231/NC or shTLN1 cells were injected via tail vein of NOD/SCID mice (n = 8 per group). After 8 weeks, lung tissue was removed to evaluate the numbers and sizes of lung metastatic nodules by analysis of lung sections.

For the evaluation of treatment with C67399, after inoculation with MDA-MB-231 cells, the mice were randomized and intravenously treated with vehicle (5% DMSO in PBS) or 1.75 mg/kg C67399 twice per week for 3 weeks. Tumour growth was monitored by diameter for up to 21 days post-inoculation, at which point the mice were sacrificed. The tumours were recovered, and their volume and weight were measured (n = 5 per group). In addition, the numbers and sizes of lung metastatic tumour nodules in the individual mice were examined by analysis of lung sections (n = 5 per group).

## Histology

Each xenograft tissue samples fixed in 10% formaldehyde solution (pH 7.0) and paraffin-embedded. Paraffin-embedded tissue sections (4 µm) were dewaxed in xylene (3 × 10 min), rehydrated through a series of graded alcohols (100%, 95%, 85%, and 75%) to water. For histology, samples were stained with haematoxylin and eosin (H&E, Servicebio, Wuhan, China). The sections were imaged under a light microscope (Olympus, Tokyo, Japan) and independently examined by two pathologists in a blinded manner.

## Statistical analysis

Data are expressed as the mean ± standard error of the mean (SEM). Differences among more than two groups were analysed by one-way ANOVA with a post hoc Newman-Keuls test, and differences between two groups were analysed by Student's t test. Disease-free survival (DFS) in each group of patients was estimated by the Kaplan-Meier method and analysed by the log-rank test. All statistical analyses were performed using SPSS 23.0 (IBM, Armonk, NY). A p-value < 0.05 was considered statistically significant.

## Results

### TLN1 overexpression is associated with poor survival in TNBC patients

Cell-cell (CC) and CEM adhesion are major structural components of the tumour microenvironment and induce a network of signals in refractory cancers, such as TNBC (*Arroyo-Crespo et al., 2019*; *Xu et al., 2012*; *Zhao et al., 2014*). Integrin family members are known as key gears in CC and CEM for metastasis of multiple cancer types (*Arroyo-Crespo et al., 2019*; *Xu et al., 2012*; *Zhao et al., 2014*).

However, we found no correlation between integrin family members and different breast cancer types using TCGA database (*Figure 1A*). Correlation between TLN1 and the integrin family in different types of the TCGA-BRCA dataset from TIMER database showed that TLN1 may play an important role in breast cancer, especially in TNBC (*Figure 1—figure supplement 1A-B*). Due to the lack of an effective target for TNBC, we switched our focus on TLN1, a partner of integrins in TNBC.

To evaluate the clinical significance of TLN1 in TNBC, breast cancer tissues and adjacent non-tumour breast tissues from 171 surgically resected TNBC patients were studied. Immunohistochemistry of TLN1 showed that TLN1 was ubiquitously expressed in the membrane and cytoplasm of paraneoplastic and tumour tissues (*Figure 1B*). Western blotting of freshly dissected TNBC tumours revealed that the relative protein levels of TLN1 in TNBC tumour tissues were significantly higher than those in para-cancerous breast tissues (*Figure 1C*). Statistical analysis shows that TLN1 was highly expressed in 43.9% of TNBCs, and the DFS of high and low expression of TLN1 shows higher expression of TLN1 was significantly associated with shorter DFS in TNBCs (HR = 3.19, p < 0.001, *Figure 1D*). Stratified analysis indicated that high level of TLN1 expression were significantly associated with T3-T4 stage (p = 0.03), positive lymph node metastasis (p < 0.001), and high Ki67 index (p < 0.001) in all 171 TNBCs (*Figure 1E*). Besides, high expression of TLN1 was detected in chest wall recurrence, lymphatic metastasis, and intestinal metastasis (*Figure 1B*). Hence, TLN1 upregulation was associated with poor prognosis in TNBC patients.

Considering that TLN2 shares 76% protein sequence identity with TLN1 (*Gough and Goult, 2018*), we analysed the expression of *TLN2* in the breast cancer dataset from TCGA database. Results showed that *TLN2* mRNA in breast cancer tissue were significantly lower than in normal breast tissues (p < 0.01), and there was no difference in levels between luminal and TNBC, HER2+ and TNBC subtypes, except for the comparison between luminal and HER2+ subtypes (p = 0.006) (*Figure 1—figure supplement 1C*). Also, there was no significant difference in survival between high and low expression of *TLN2* within each subtype of breast cancer (*Figure 1—figure supplement 1D*). Therefore, our subsequent study mainly focused on TLN1 rather than TLN2.

## Depletion of TLN1 inhibits the growth of TNBC cells in vitro and in vivo

To understand the role of TLN1 in different types of breast cancer, relative protein levels of different types of breast cancer cell lines were detected by western blot. The TNBC cell lines MDA-MB-231 and BT-549 showed much higher levels of TLN1 than luminal cell line MCF-7 and HER2+ cell line SK-BR-3 cells (*Figure 2A*). We then employed shRNA-based lentivirus technology to generate MDA-MB-231 cells with stable knockdown of TLN1 expression (MDA-MB-231/shTLN1 cells), in which TLN1 expression was obviously reduced compared to negative control cells, named as MDA-MB-231/NC (*Figure 2B*). Silencing *TLN1* significantly inhibited the proliferation of MDA-MB-231 cells (*Figure 2C*) and increased the frequency of spontaneous apoptosis (*Figure 2D*) and the number of apoptosis-related vacuoles in MDA-MB-231 cells (*Figure 2E*). More importantly, silencing *TLN1* was shown significant reduction of tumour size and weight in NOD/SCID mice implanted with $10^6$ MDA-MB-231/NC or shTLN1 cells in the mammary fat pad (*Figure 2F*). Together, these data demonstrate that silencing *TLN1* can inhibit the growth of TNBC cells in vitro and in vivo.

## Depletion of TLN1 inhibits the adhesion and metastasis of TNBC cells in vitro and in vivo

Since TLN1 is crucial for tumour cell adhesion and metastasis (*Zhang et al., 2011*), we tested the effect of silencing *TLN1* on MDA-MB-231 cell adhesion, migration, and invasion in vitro. We found that silencing *TLN1* significantly inhibited the cell adhesion, migration, and invasion of MDA-MB-231 cells (*Figure 3A and B*) and BT549 cells (*Figure 3—figure supplement 1A-C*). Additionally, silencing *TLN1* significantly increased the expression of CK18 and E-cadherin but significantly decreased the expression of N-cadherin and vimentin in MDA-MB-231 cells, suggesting that silencing *TLN1* inhibited EMT in TNBC cells (*Figure 3C and D*). More importantly, silencing *TLN1* suppressed lung metastasis in vivo, as demonstrated by significant decreases in the number and size of lung metastatic nodules in mice injected with shTLN1 cells (*Figure 3E*). While overexpression of integrin β1 could rescue invasion, migration, and adhesion in shTLN1 cells (*Figure 3F–H*). Thus, silencing *TLN1* may inhibit TNBC cell adhesion and metastasis by attenuating EMT.

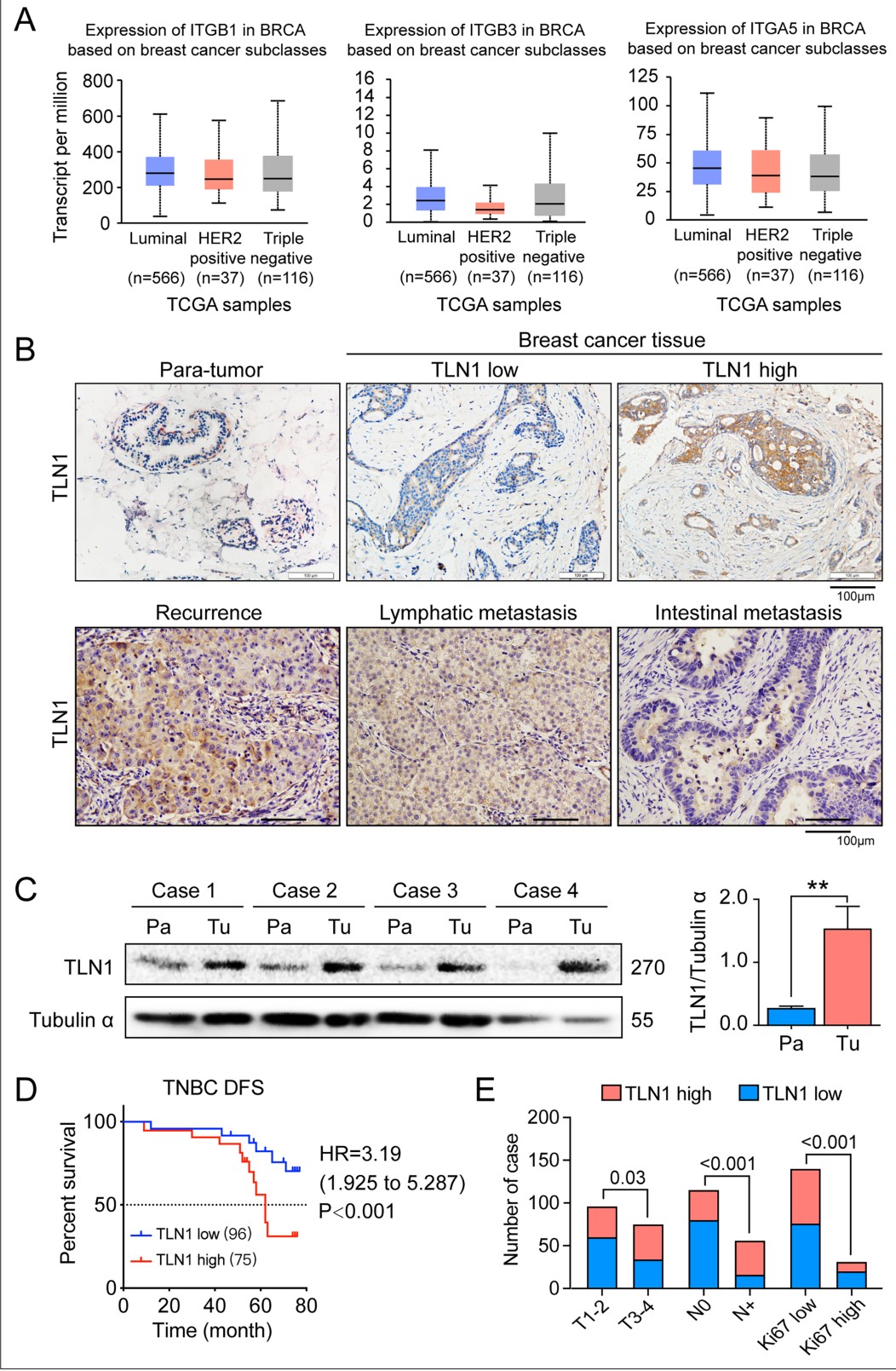

**Figure 1.** TLN1 upregulation is associated with poor disease-free survival (DFS) in triple-negative breast cancer (TNBC). (**A**) The expression of ITGB1, ITGB3, and ITGA5 in breast cancer subclasses and normal breast tissue using The Cancer Genome Atlas (TCGA) samples (n = 114 of normal samples, n = 556, 37, and 116 of luminal, human epidermal growth factor receptor 2 [HER2] positive and TNBC samples, respectively; p-value is the result

*Figure 1 continued on next page*

*Figure 1 continued*

of comparison with normal samples, respectively). (**B**) Representative immunohistochemistry images of TLN1 expression in TNBC tissue, chest wall recurrence, lymphatic metastasis, and intestinal metastasis (scale bar, 100 μm). (**C**) Western blot analysis of TLN1 expression in fresh TNBC and para-cancerous tissues (n = 4, p = 0.009). (**D**) The DFS of 171 TNBC patients in the cohort was estimated by the Kaplan-Meier method, and the difference between groups with high and low TLN1 expression was compared within each set of patients listed and analysed by log-rank analysis (HR = 3.19, p < 0.001). (**E**) Chi-square analysis of high or low TLN1 expression with T-stage, N-stage, and Ki67 index. Data are either presented as representative images or expressed as the mean ± standard error of the mean (SEM) of each group. p < 0.01 was indicated by **.

The online version of this article includes the following source data and figure supplement(s) for figure 1:

**Source data 1.** TLN1 upregulation is associated with poor DFS in TNBC.

**Figure supplement 1.** Analysis of TLN2 expression in a dataset from the The Cancer Genome Atlas (TCGA) database.

**Figure supplement 1—source data 1.** Analysis of TLN2 expression in a dataset from the TCGA database.

## Silencing *TLN1* inhibits FA formation and integrin β1 signalling in MDA-MB-231 cells

To understand the molecular mechanisms underlying the effects of silencing *TLN1*, we compared protein expression profiles between MDA-MB-231/NC and shTLN1 cells using TMT proteomics. There were 156 DEPs between these cells; 76 were upregulated, whereas 80 were downregulated (*Figure 3—figure supplement 2A*). The results of Gene Ontology (GO) functional enrichment analysis of DEPs indicated that these TLN1-related DEPs were involved in tissue development, positive regulation of cell migration, microtubule organization, cell differentiation, and protein-binding activity (*Figure 3—figure supplement 2B, C, D*).

Given that mutations within both integrins β1 (*Bouaouina et al., 2008*) and β3 (*Tadokoro et al., 2003*) may abolish TLN binding and decrease integrin affinity, TLN1 ablation universally leads to integrin adhesion defects (*Chen et al., 2017*; *Jin et al., 2015*). These experiments inspired us to verify whether TLN1 could bind integrin β1 or β3 in TNBC. Firstly, we validated the interactions between TLN1 and integrin β1 or β3 in MDA-MB-231 cells by immunoprecipitation and found that there was a significant interaction between TLN1 and integrin β1 but not between TLN1 and integrin β3, though a non-specific band was seen in the TLN1 pull-down of integrin β3, which was probably a trace left by electrophoresis (*Figure 4A*). Subsequently, immunofluorescence showed that TLN1 and integrin β1 were scattered in the cytoplasm while co-located in FAs at both poles of the cell (*Figure 4B*). As TLN1 is a critical component of FAs which can be observed by vinculin and actin-phalloidin staining (*Bennett et al., 2018*), we further observed the effect of *TLN1* silencing on FAs formation by confocal fluorescence microscopy. Results showed that silencing *TLN1* led to significantly shorter cells, fewer FAs, the predominant localization of FAs on the cell membrane, and significant thickening of the actin cortex (*Figure 4C*). Moreover, light microscopic imaging showed that the morphology of shTLN1 cells was no longer as fusiform as MDA-MB-231/NC cells (*Figure 4D*). Western blot analysis further indicated that silencing *TLN1* in MDA-MB-231 cells reduced the relative levels of integrin β1, as well as the levels of phosphorylated AKT and FAK (*Figure 4E*). These data indicate that silencing *TLN1* attenuates integrin β1 signalling in TNBC cells, which results in impaired dynamic formation and maturation of FAs.

## C67399 is found to block the integrin β1 binding site of TLN1 and reduces the malignant behaviours of TNBC in vitro

Since we showed that TLN1/integrin β1 signalling is crucial in malignant behaviours of TNBC cells, we next screened small molecules from Enamine database that could block the binding of TLN1 and integrin β1, based on a novel CSTPPI using our in-house developed FIPSDock software (*Liu et al., 2013*; *Figure 5—figure supplement 1A*). In brief, the conformational ensembles of TLN1 and integrin β1 were first generated via 20 ns molecule dynamics simulation. Subsequently, the dynamic binding pockets of TLN1 and integrin β1 were used for further computational screening. Structurally, the TLN1 'head' is comprised largely of an FERM domain, which contains F0, F1, F2, and F3 domains (band 4.1, ezrin, radixin, and moesin) (PDB ID: 3IVF) (*Goult et al., 2013*). As the S1 and S2 chains of the PTB F3 domain are crucial for integrin binding and activation, the F3 domain of TLN1 was targeted

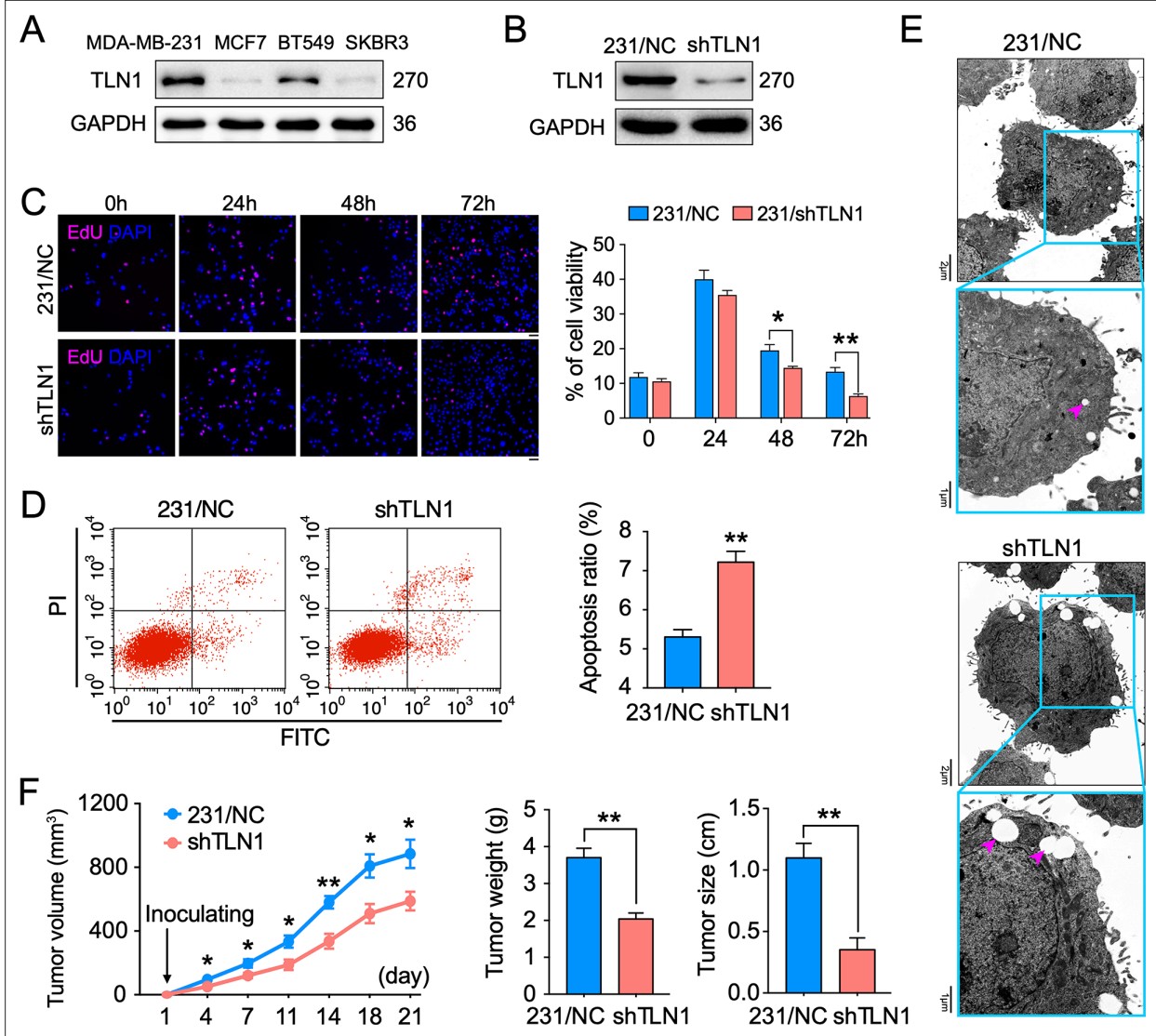

**Figure 2.** Silencing *TLN1* promotes apoptosis and inhibits tumour growth of MDA-MB-231 cells. (**A**) TLN1 expression in the indicated breast cancer cell lines, using western blotting. (**B**) The efficiency of *TLN1* silencing with shRNA in MDA-MB-231 cells was evaluated by western blotting. (**C**) Silencing *TLN1* inhibited the proliferation of MDA-MB-231 cells (scale bar, 20 µm, p < 0.05 at 48 hr and p < 0.01 at 72 hr). (**D**) Silencing *TLN1* enhanced spontaneous apoptosis in MDA-MB-231 cells, which was detected by Flowcyto. (**E**) Silencing *TLN1* increased apoptosis-related vacuoles (red arrows), which was detected by transmission electron microscopy (TEM). (**F**) Silencing *TLN1* reduced the growth of xenografted triple-negative breast cancer (TNBC) tumours in NOD/SCID mice (n = 8 per group). Data are presented as representative images, flow cytometric plots, or the mean ± standard error of the mean (SEM) of each group from three separate experiments. *p < 0.05, **p < 0.01 vs. the NC group.

The online version of this article includes the following source data for figure 2:

**Source data 1.** Silencing TLN1 promotes apoptosis and inhibits tumour growth of MDA-MB-231 cells.

---

(*Figure 5A*). Based on the scores of the corresponding docking conformation and the pocket mode of action, we firstly selected the top eight small molecular compounds (named C1-C8) with the highest affinity score to test the drug sensitivity in vitro (*Figure 5—figure supplement 1B*). Due to the poor solubility of C1, C3, C5, C7, and C8, we only verified the drug sensitivity of the remaining C2, C4, and C6 (C67399) compounds to MDA-MB-231 cells. The results showed that C67399 ($C_{38}H_{48}O_5$, 584.79 Da, 4903–2135, Chemdiv Compond, San Diego, CA) could obviously inhibit the cell viability of MDA-MB-231 cells (*Figure 5—figure supplement 1C*). We also found that C67399 could stably bound TLN1 with a low dissociation equilibrium (*Figure 5B*). Further MD simulations revealed that C67399 interacts with Thr354, Ala389, Gln390, Ala393, Ile396, Asp397, and Ile398, in addition to Trp359, in the hydrophobic pocket of the F3 domain of TLN1 (*Figure 5C*). Moreover, the total interaction energy

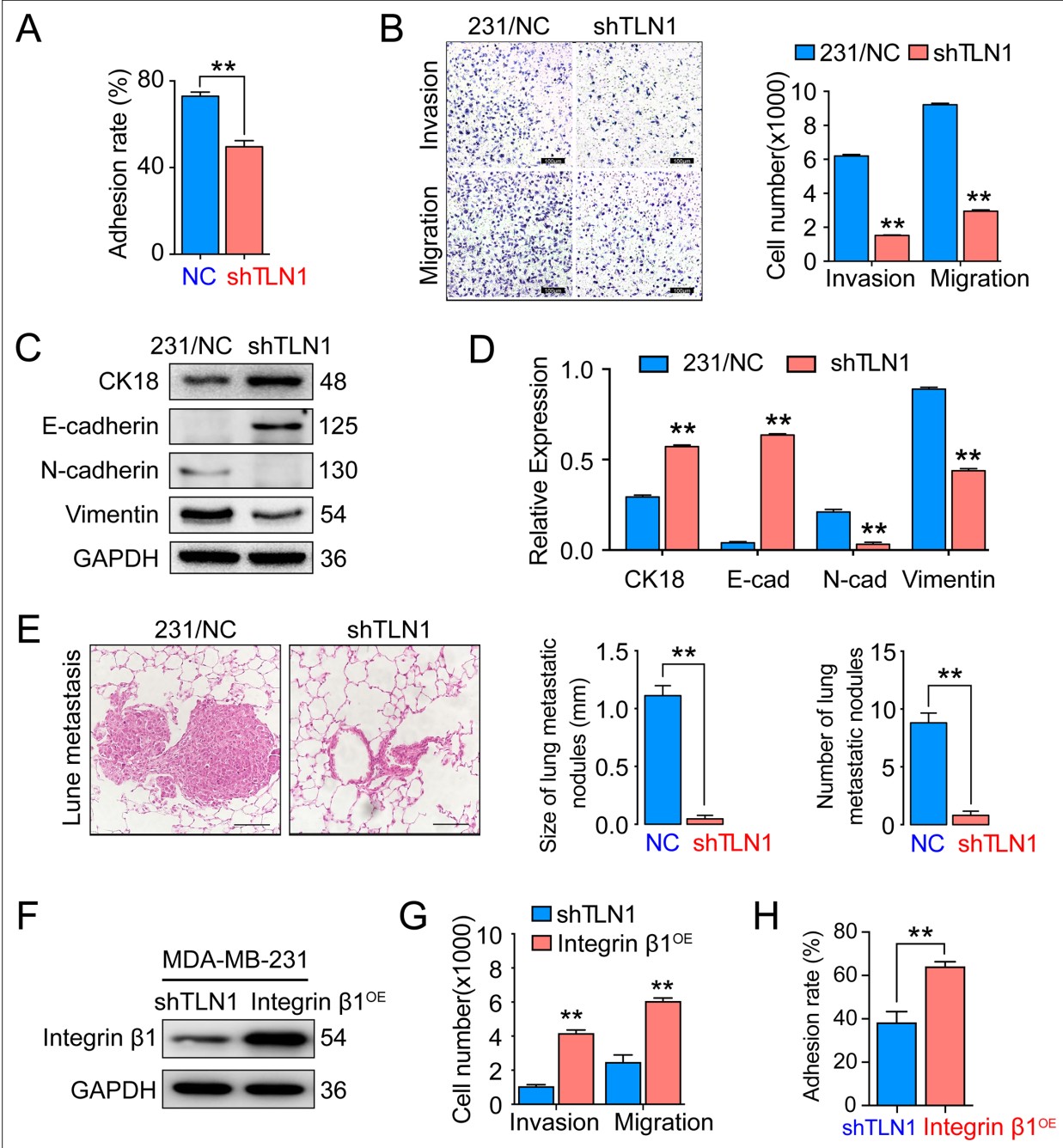

**Figure 3.** Silencing *TLN1* reduces triple-negative breast cancer (TNBC) cell motility by blocking epithelial-mesenchymal transformation (EMT). (**A**) *TLN1* silencing decreased the adhesion of MDA-MB-231 cells (n = 3, p < 0.01). (**B**) Transwell assay was used to evaluate the migration and invasion of MDA-MB-231/NC and shTLN1 cells (n = 3, p < 0.01; scale bar, 100 μm). (**C and D**) The relative expression level of CK18, E-cadherin, N-cadherin, and vimentin relative to those of GAPDH in MDA-MB-231/NC and shTLN1 cells, using western blotting (n = 3, p < 0.01, respectively). (**E**) Haematoxylin and eosin (H&E) staining of mitigated lung metastasis derived from MDA-MB-231/NC and shTLN1 tumours in NOD/SCID mice (n = 5–8 per group; scale bar, 50 μm). (**F**) Western blot of integrin β1 overexpression in MDA-MB-231/shTLN1 cells. (**G**) Transwell analysis of invasion and migration following integrin β1 overexpression. (**H**) Adhesion assay after integrin β1 overexpression. Data are representative images or expressed as the mean ± standard error of the mean (SEM) of each group from three separate experiments for in vitro and 5–8 per group for in vivo studies. *p < 0.05, **p < 0.01 vs. the control group.

The online version of this article includes the following source data and figure supplement(s) for figure 3:

**Source data 1.** Silencing TLN1 reduces TNBC cell motility by blocking EMT.

**Figure supplement 1.** Silencing *TLN1* in BT549 cell line reduces adhesion, invasion, and migration.

*Figure 3 continued on next page*

*Figure 3 continued*

**Figure supplement 1—source data 1.** Silencing TLN1 in BT549 cell line reduces adhesion, invasion and migration.

**Figure supplement 2.** Tandem mass tag analysis of differentially expressed proteins (DEPs) between MDA-MB-231/NC and shTLN1 cells.

**Figure supplement 2—source data 1.** Tandem mass tag analysis of differentially expressed proteins (DEPs) between MDA-MB-231/NC and shTLN1 cells.

spectrum shows that the addition of C67399 reduces the energy expenditure between TLN1 and integrin β1 (*Figure 5D*). These findings indicated that small-molecule C67399 could block the combination of TLN1 and integrin β1 from the point of view of protein structure and in silico modelling.

Next, we analysed the effect of C67399 on TNBC cells. Dose-response curve for C67399 in MDA-MB-231 cells revealed that the half maximal inhibitory concentration (IC50) of C67399 was 2.0 μM (*Figure 5E*). Functionally, treatment with C67399 significantly decreased the viability, adhesion, migration, and invasion of MDA-MB-231 cells and BT549 cells (*Figure 5F, G and H* and *Figure 5—figure supplement 1D-E*). More importantly, 2.0 μM C67399 significantly reduced the expression of integrin β1, AKT, FAK, and phosphorylated FAK in MDA-MB-231 cells, while did not affect the expression of integrin β3 (*Figure 5I*). Additionally, immunoprecipitation results revealed that C67399 inhibited the binding of TLN1 to integrin β1 in MDA-MB-231 cells (*Figure 5J*). Our findings suggest that targeting TLN1 with C67399 suppresses TNBC cell malignancy in vitro.

## C67399 inhibits the tumour growth and metastasis of MDA-MB-231 cells in mice

To better understand the therapeutic effect of C67399 on TNBC in vivo, MDA-MB-231 cells were injected into the fat pad or tail vein of NOD/SCID mice with or without C67399 treatment. Mice that received intravenous administration of 1.75 mg/kg C67399 twice per week showed significant reductions in the tumour volume (*Figure 6A and B*), the number, and size of lung metastatic nodules (*Figure 6C, D and E*), indicating decreased tumour growth and attenuated lung metastasis of implanted shTLN1 cells. Furthermore, a decreased proliferation index of Ki67 was observed in the C67399-treated xenograft tumours using immunohistochemistry, compared to the untreated MDA-MB-231 tumours (*Figure 6F*).

We also performed H&E staining of other organs harvested from the mice treated with C67399, and no structural changes suggestive of toxicity were observed in the heart, liver, spleen, lungs, kidneys, or other organs (*Figure 6—figure supplement 1A*), as well as no significant differences in Alcian blue staining of intestine, alanine aminotransferase, and aspartate aminotransferase activities in mice serum and blood cell counts (*Figure 6—figure supplement 1B-D*). These results suggested that C67399 could inhibit the tumour growth and metastasis of MDA-MB-231 cells in vivo, without causing obvious structural toxic changes.

Collectively, our findings indicated that TLN1 can bind and activate integrin β1 in TNBC cells. On the one hand, it can regulate the dynamic formation and maturation of FAs, induce EMT to promote tumour metastasis; on the other hand, it facilitates tumour growth by inhibiting apoptosis. Fortunately, C67399 can inhibit the binding of TLN1 to integrin β1, as well as a series of integrin β1-related pathways, and ultimately inhibit the malignancy of TNBC (*Figure 6G*).

## Discussion

TNBC is a highly aggressive subtype of breast cancer characterized by rapid proliferation, early invasion, recurrence, and metastasis (*Mostert et al., 2015*). TLN1 is an integrin-activated, tension-sensitive FA component that directly links integrins in the plasma membrane to myosin cytoskeleton (*Haage et al., 2018*; *Jin et al., 2015*). Beyond the FA component, TLN1 is also associated with chemosensitivity to cancer therapy. Loss of TLN1 function was reported to significantly enhance chemosensitivity in TNBC cell lines, but not in hormone positive cell lines (*Singel et al., 2013*). They concluded that TLN1 is a regulator of response to docetaxel and a potential therapeutic target for TNBC, but not in other types of breast cancer (*Singel et al., 2013*). In addition to its role in chemosensitivity, we found that TLN1 has a carcinogenic effect independent of chemosensitivity in TNBC, and TLN1 initiates the growth and migration of TNBC. Therefore, we developed a specific molecule that more effectively

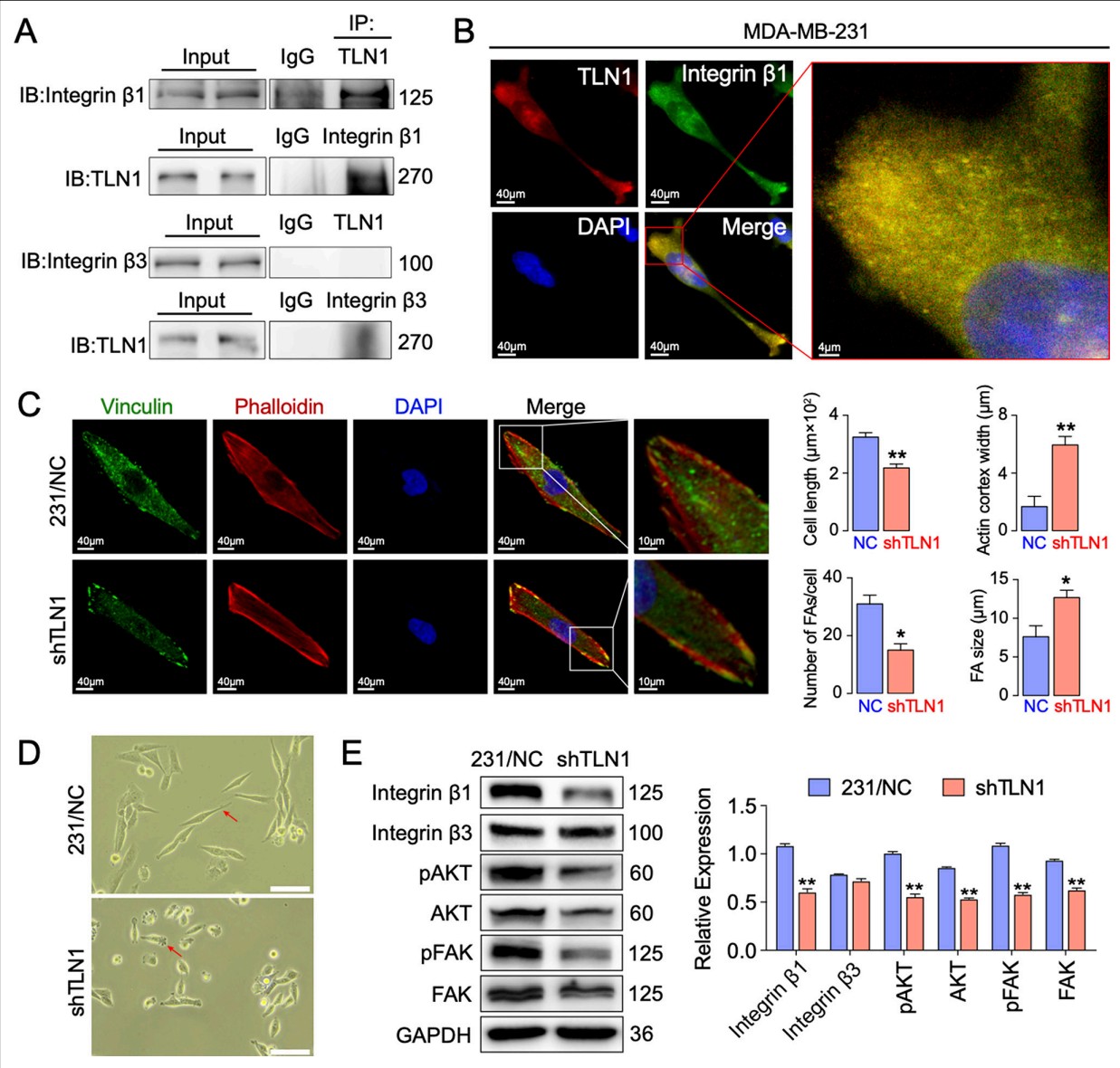

**Figure 4.** Silencing *TLN1* reduces focal adhesion (FA) dynamic formation and integrin β1-mediated signalling in MDA-MB-231 cells via loss of interactions with integrin β1. (**A**) Immunoprecipitation analysis of the interaction of TLN1 with integrin β1 or integrin β3 in MDA-MB-231 cells. (**B**) Immunofluorescence images of TLN1 and integrin β1 in MDA-MB-231 cells (scale bar, 40 µm; 5 µm for enlarged image). (**C**) Immunofluorescence confocal microscopy analysis of FAs and the actin cortex in MDA-MB-231/NC and shTLN1 cells (scale bar, 40 µm; 10 µm for enlarged image). (**D**) Light microscopy images of MDA-MB-231/NC and shTLN1 cells (scale bar, 100 µm). (**E**) Western blot analysis of the relative levels of integrin β1, integrin β3, total AKT, total FAK, phosphorylated AKT (Ser473), and phosphorylated FAK (Tyr397) in MDA-MB-231/NC and shTLN1 cells. GAPDH used as the control to evaluate relative expression. Data are presented as representative images or the mean ± standard error of the mean (SEM) of each group from three separate experiments. **p < 0.01 vs. the NC group.

The online version of this article includes the following source data for figure 4:

**Source data 1.** Silencing TLN1 reduces FA dynamic formation and integrin β1-mediated signalling in MDA-MB-231 cells via loss of interactions with integrin β1.

inhibits the oncogenic effect of TLN1, which may be developed as a promising clinical treatment for TNBC, while potentially increasing chemotherapy sensitivity.

In this study, we revealed that high expression of TLN1 in TNBC was associated with poor prognosis and malignant behaviour, including proliferation, cell adhesion, EMT, invasion, and migration. These data indicate that TLN1 promotes TNBC malignancy as an oncogenic gene, which is in accordance with previous observations in prostate cancer, colon cancer, and oral squamous cell carcinoma

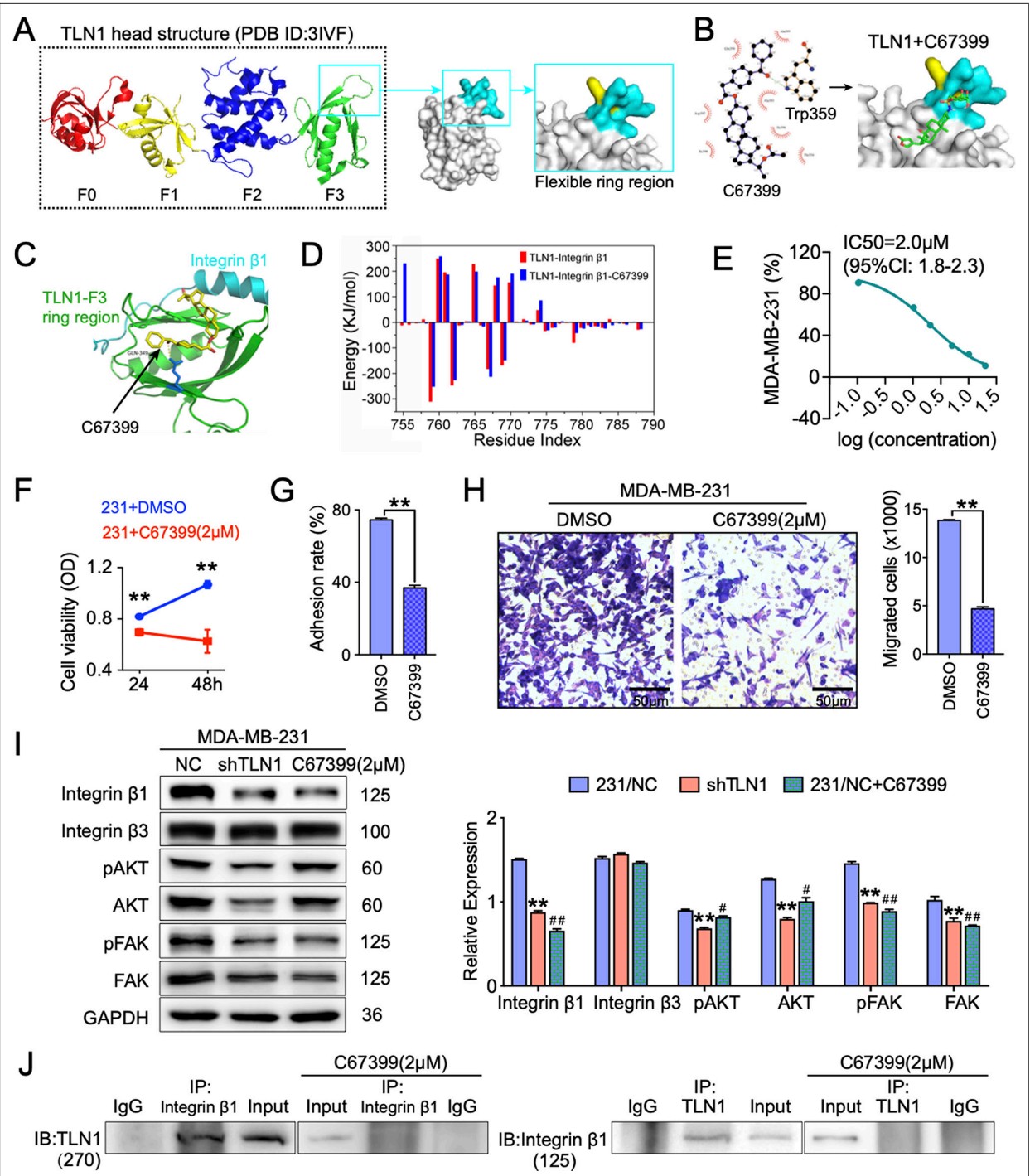

**Figure 5.** C67399 blocks TLN1-integrin β1 binding and attenuates the malignant behaviours of MDA-MB-231 cells. (**A**) Complete structure of the human TLN1 'head' with the F0, F1, F2, and F3 domains, with the predicted interaction between the TLN1 F3 domain (PDB ID: 3IVF) and integrin β1 at far right. The targeting of the flexible ring region of the F3 domain of the TLN1 head structure was labelled in cartoon mode and the head structure was labelled in surface mode using PyMOL software. The yellow- and paon-labelled regions indicate the hydrophobic ring structure of the F3 domain of TLN1, with the yellow-labelled region representing the K324 residue. (**B**) Docking mode diagram of the chemical molecule C67399 and TLN1; the hydrophobic pocket formed by the flexible ring of C67399 and the F3 domain was screened by FIPSDock. (**C**) The main favourable contributions to the binding of C67399 came from hydrophobic contacts with Thr354, Ala389, Gln390, Ala393, Ile396, Asp397, and Ile398, using a closer analysis of the complex structure and energy term. (**D**) Molecular dynamic simulations of the electrostatic interactions between C67399 and Trp359. (**E**) The dose-response curve of C67399 in MDA-MB-231 cells using CCK-8 kit (IC50 = 2.0 μM). (**F–H**) C67399 treatment significantly reduced the viability(F), adhesion(G), and migration(H) of MDA-MB-231 cells. (**I**) Western blot analysis of the relative levels of integrin β1, integrin β3, total AKT, total FAK, phosphorylated AKT

*Figure 5 continued on next page*

*Figure 5 continued*

(Ser473), and phosphorylated FAK (Tyr397) in MDA-MB-231/NC and shTLN1 cells, as well as in MDA-MB-231/NC treated with C67399 (2 μM for 48 hr). GAPDH used as the control to evaluate relative expression. (J) Immunoprecipitation analysis of TLN1 and integrin β1 in the presence or absence of C67399 treatment (2 μM for 48 hr). Data are shown as representative images, charts, or the mean ± standard error of the mean (SEM) of each group from three separate experiments. #p < 0.05, ##p < 0.01, and **p < 0.01 vs. the NC group.

The online version of this article includes the following source data and figure supplement(s) for figure 5:

**Source data 1.** C67399 blocks TLN1–integrin β1 binding and attenuates the malignant behaviours of MDA-MB-231 cells.

**Figure supplement 1.** The process of computational screening small-molecule compounds targeting TLN1.

**Figure supplement 1—source data 1.** The process of computational screening small molecule compounds targeting TLN1.

---

(*Bostanci et al., 2014*; *Jin et al., 2015*; *Lai et al., 2011*). Given that it is very challenging to predict survival for TNBC patients after surgical resection (*Brockwell et al., 2019*), our data indicated that higher TLN1 expression could be a prognostic factor for TNBC patients. However, further validation of these findings is needed.

Our findings also highlight that TLN1 is a potential therapeutic target for TNBC. Specifically, TLN1 expression in TNBC cells is much higher than in other subtypes of breast cancer cell lines, and silencing *TLN1* significantly attenuated the malignancy of MDA-MB-231 cells, reducing TNBC tumour growth and lung metastasis. Mechanically, it is hypothesized to inhibit cancer metastasis through suppressing EMT. These data were consistent with previous observations (*Thapa et al., 2017*) and support the notion that TLN1-related signalling is crucial for TNBC malignancy.

It is well known that malignant TNBC cells can adhere to the ECM by activating integrins (*Bays and DeMali, 2017*; *Bosch-Fortea and Martín-Belmonte, 2018*; *Hynes, 2002*; *Lock et al., 2008*) and the activation of integrin-related signalling promotes EMT and metastasis in TNBC (*Wen et al., 2019*). Major insights into TLN1 demonstrated that its 'head' can bind the cytoplasmic tails of integrin β through its PTB domain, resulting in integrin β activation (*Tadokoro et al., 2003*). This activation links integrins to the cytoskeleton (*Critchley and Gingras, 2008*) by activating FAK and Src and further promotes downstream PI3K/AKT signal transduction to promote cancer malignancy (*Jin et al., 2015*; *Klapholz and Brown, 2017*; *Sun et al., 2019*). In our results, TLN1 interacts with integrin β1 and silencing *TLN1* alters many proteins expression and significantly reduces FAs dynamic formation in MDA-MB-231 cells, which is similar to observations in fibroblasts and endothelial cells (*Kopp et al., 2010*; *Nader et al., 2016*). In addition, silencing *TLN1* was found to significantly reduce integrin β1 levels as well as the levels of phosphorylated FAK. The reduction in integrin might be a result of *TLN1* silencing-mediated integrin degradation (*Chinthalapudi et al., 2018*). These data suggest that TLN1 is crucial for FA dynamic formation, adhesion, and invasion in TNBC cells, as well as tumour growth and metastasis. Additionally, silencing *TLN1* significantly attenuated AKT phosphorylation in MDA-MB-231 cells, demonstrating the mitigation of TLN1/integrin β1 signalling in TNBC cells.

The actin cortex is a thin layer of filamentous actin that lies beneath the plasma membrane (*Litschko et al., 2019*). During EMT, actin reorganizes from a mostly cortical arrangement into stress fibres and lamellipodia (*Chalut and Paluch, 2016*). To the best of our knowledge, this study provides the first evidence that silencing *TLN1* induces thickening of the actin cortex in TNBC cells, which may initiate cell surface contractile tension and result in local contractions and drive cell deformations (*Chugh et al., 2017*). Our findings indicated that loss of TLN1 led to actin redistribution, resulting in changes in membrane dynamics, and this loss may interfere with the efficient assembly or turnover of FAs, which is required for directional migration (*Garcin and Straube, 2019*). Therefore, TLN1 may regulate the dynamic formation of FAs by reducing cortical actin, and then promote cell migration in TNBC cells.

In this study, for the first time, we identified a novel small-molecule compound, C67399, which blocks TLN1 binding to integrin β1, through a novel computational approach named as CSTPPI against the binding interface of TLN1 with integrin β1. The advantage of this CSTPPI is that conformational ensembles of TLN1-integrin β1 interfaces were used as targets for the computational screening via our in-housed developed docking tool. Therefore, the flexibility of protein-protein interface was given full consideration in our CSTPPI to improve the accuracy. This is the first study of identifying small-molecule compounds by targeting dynamic protein-protein binding interface and this may be extended to other PPI study. Remarkably, we found that C67399 could significantly ablate the

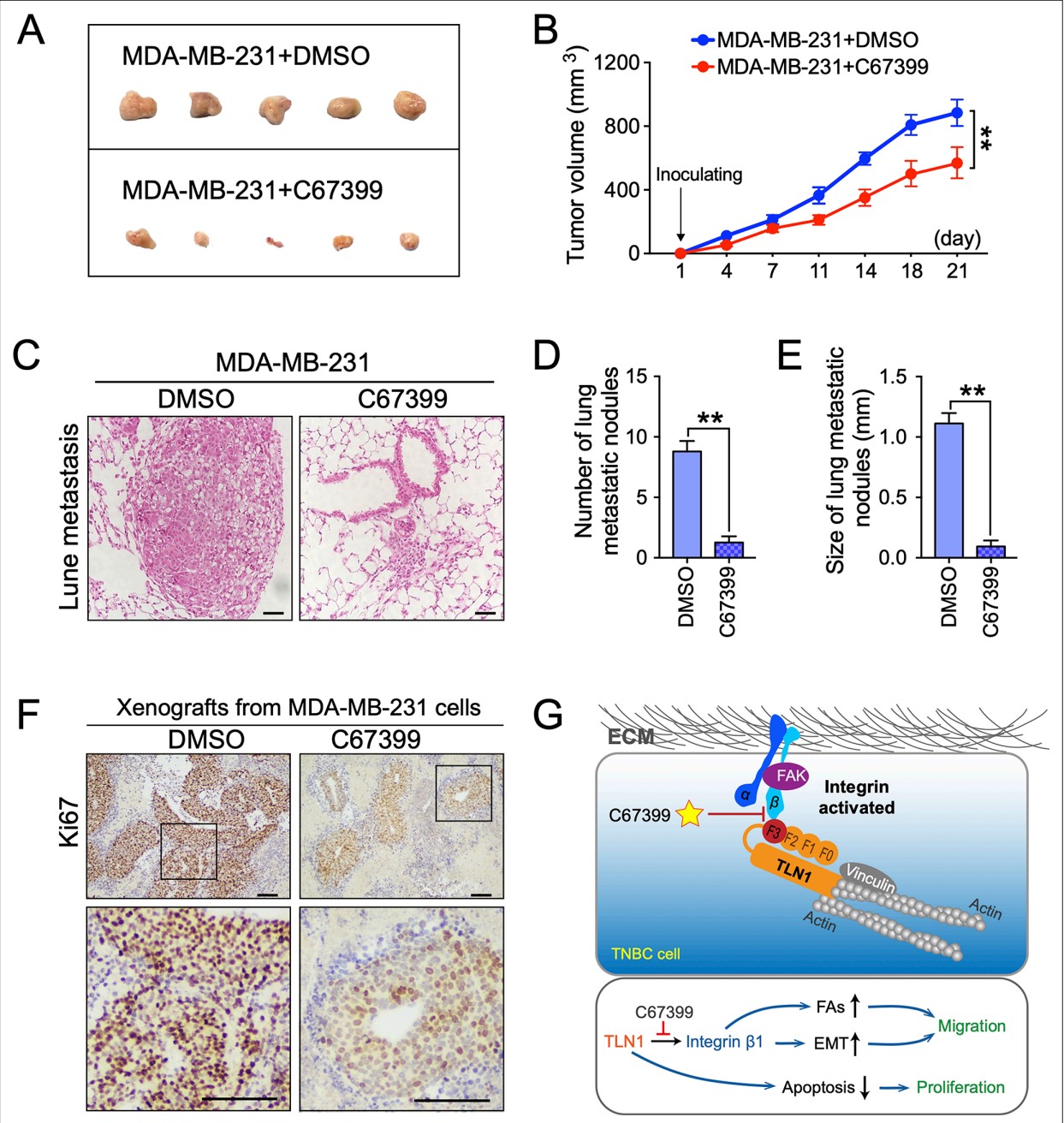

**Figure 6.** C67399 inhibits the growth and metastasis of implanted triple-negative breast cancer (TNBC) cells. MDA-MB-231 cells were injected into fat pad or tail vein of NOD/SCID mice to establish tumour model. Mice were intravenously injected with 1.75 mg/kg C67399, twice a week for 3 weeks. (**A and B**) The tumour volume of xenografts derived from MDA-MB-231 cells treated with or without C67399 (n = 5–8 per group, p < 0.01). (**C–E**) The number and size of lung metastatic tumour nodules in mice of MDA-MB-231 cells with or without C67399 treatment (n = 5–8 per group, p < 0.01). (**F**) Immunohistochemical staining for Ki67 of xenografted tumours derived from MDA-MB-231 cells in the presence or absence of C67399 treatment. (**G**) Diagram illustrating the function of TLN1 in TNBC. TLN1 could bind and activate integrin β1 in TNBC cells. On the one hand, it can regulate the dynamic formation and maturation of focal adhesions (FAs), induce epithelial-mesenchymal transformation (EMT), and promote tumour metastasis; on the other hand, it can promote tumour proliferation by inhibiting apoptosis. A small-molecule C67399 was developed to inhibit the binding of TLN1 to integrin β1, as well as a series of integrin β1-related pathways, and ultimately inhibit the malignancy of TNBC.

The online version of this article includes the following source data and figure supplement(s) for figure 6:

**Source data 1.** C67399 inhibits the growth and metastasis of implanted TNBC cells.

**Figure supplement 1.** Toxicity of C67399 in mice.

**Figure supplement 1—source data 1.** Toxicity of C67399 in mice.

invasiveness of MDA-MB-231 cells and suppress the lung metastasis in vivo. These novel findings suggest that TLN1 is a promising therapeutic target and C67399 is a valuable candidate compound for TNBC intervention. C67399 is the first small-molecule inhibitor of its kind that interfere in the binding interface of TLN1 with integrin β1.

In summary, our data suggest that TLN1 is significantly upregulated in TNBC cells and TLN1 interacts with integrin β1 to activate downstream signalling pathways including PI3K/AKT and FAK pathways. Furthermore, the interaction of TLN1 with integrin β1 facilitates the formation of FAs and cell adhesion, leading to the malignancy and metastasis of TNBC by enhancing EMT. Hence, TLN1 could be a valuable prognostic factor and a promising therapeutic target for TNBC. Moreover, we identified a small-molecule compound C67399 via computational screening and we demonstrated that genetically downregulating *TLN1* expression or blocking the binding of TLN1 with integrin β1 by C67399 could inhibit the aggressiveness and metastasis of TNBC.

## Additional information

### Competing interests
Caigang Liu: Reviewing editor, *eLife*. The other authors declare that no competing interests exist.

### Funding

| Funder | Grant reference number | Author |
|---|---|---|
| National Natural Science Foundation of China | 81872159 | Caigang Liu |
| Liaoning Colleges Innovative Talent Support Program | Cancer Stem Cell Origin and Biological Behavior | Caigang Liu |
| Outstanding Scientific Fund of Shengjing Hospital | 201803 | Caigang Liu |
| Outstanding Young Scholars of Liaoning Province | 2019-YQ-10 | Caigang Liu |
| National Natural Science Foundation of China | 81902607 | Yixiao Zhang |
| National Natural Science Foundation of China | 81874301 | Yongliang Yang |

The funders had no role in study design, data collection and interpretation, or the decision to submit the work for publication.

### Author contributions
Yixiao Zhang, Data curation, Formal analysis; Lisha Sun, Formal analysis, Methodology, Writing - review and editing; Haonan Li, Methodology, Software; Liping Ai, Investigation, Methodology; Qingtian Ma, Methodology, Resources; Xinbo Qiao, Software; Jie Yang, Hao Zhang, Resources; Xunyan Ou, Methodology; Yining Wang, Guanglei Chen, Formal analysis; Jinqi Xue, Xudong Zhu, Validation; Yu Zhao, Visualization, Writing - review and editing; Yongliang Yang, Project administration, Validation; Caigang Liu, Conceptualization, Funding acquisition, Project administration, Writing - review and editing

### Author ORCIDs
Lisha Sun (iD) https://orcid.org/0000-0002-4095-5026
Xinbo Qiao (iD) https://orcid.org/0000-0002-6759-921X
Caigang Liu (iD) https://orcid.org/0000-0003-3729-2839

### Ethics
Written informed consent was obtained from all the patients, and this study was approved by the institutional research ethics committee of China Medical University.
The current study was approved by the institutional research ethics committee of Shengjing Hospital of China Medical University (Project identification code: 2018PS304K, date on 03/05/2018), and each

participant signed an informed consent before being included in the study. Meanwhile, this study was performed in very strict accordance with the recommendations in the Guide for the Care and Use of Laboratory Animals of the National Institutes of Health. All surgery was performed under sodium pentobarbital anesthesia, and every effort was made to minimize suffering of the animals, and all the animals were handled according to approved Animal Ethics and Experimentation Committee protocols of Shengjing Hospital of China Medical University (Project identification code: 2018PS312K, date on 03/05/2018).

## Decision letter and Author response

Decision letter https://doi.org/10.7554/eLife.68481.sa1
Author response https://doi.org/10.7554/eLife.68481.sa2

## Additional files

### Supplementary files
Transparent reporting form

### Data availability
All data generated or analysed during this study are included in the manuscript and supporting files.

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
