## [Editor Report]

The paper is of interest to preclinical and translational scientists in the field of breast cancer. It details the identification, characterization and selection of a novel drug candidate, based on the biology of a select target gene, with the premise to rationalize a new therapy for triple-negative breast cancer. The data presented support the proposed hypotheses, and the conclusions are well supported by the results.

---

## [Decision Letter]

**Decision letter after peer review:**

Thank you for submitting your article "Binding blockade between TLN1 and integrin β1 represses metastasis of triple-negative breast cancer" for consideration by *eLife*. Your article has been reviewed by 3 peer reviewers, and the evaluation has been overseen by a Reviewing Editor and Mone Zaidi as the Senior Editor. The following individual involved in review of your submission has agreed to reveal their identity: Bedrich Eckhardt (Reviewer #3).

Essential revisions:

1) Referees comments must be addressed in full.

2) New data are required before re-submission, and the manuscript will be returned to the referees if a revised version is submitted.

*Reviewer #1:*

The authors established TLN1 as a potential therapeutic target for triple-negative breast cancer, an aggressive subtype, with very few treatment options. The authors used IHC on patient samples and concluded that high TLN1, but not TLN2 and other integrins expression is an independent prognostic marker for BrCa. It is important to note that TLN2/integrin expression was studied in TCGA dataset and not on the same set of samples and the conclusions on TLN2/integrins should be re-considered. The authors presented strong data on effect of TLN downregulation on cell proliferation, migration, invasion, stemness, EMT and in vivo tumor growth and metastasis. Their data suggest all aspects of tumor progression are impacted by TLN1 downregulation. Mechanistically, TLN1 downregulation impacted LATS/YAP/TAZ pathway as well as EMT proteins. The authors concluded that TLN1 silencing reduced tumor growth through Hippo/Yap signaling, which is not justified as there was no data presented, where activation of YAP/TAZ rescued stemness/tumor growth in TLN1 ablated cells. Presented data is just correlational. Same is true regarding their conclusion on EMT and Metastasis. Next, the authors confirmed the effect of TLN1 alteration on focal adhesions formation as well as other integrin signaling pathways. In the last part of the manuscript, the authors described the development and characterization of small molecule, which can inhibit interaction between beta1 integrin and TLN1. This inhibitor, C67399, affected viability, adhesion, migration, tumor growth and metastasis. However, it is important to point that all these effect on malignancy can be explained by the viability. Since cells were less viable, it is likely that non-viable cells will not adhere, migrate, form tumor or mets. This is an interesting article, but text needs to be re-worded to justify conclusions based on the data presented.

Recommendations for the authors:

1. Figure 1: To invalidate the role of TLN2/integrins, the authors should either use same patient samples with IHC for TLN2 or use TCGA for TLN1 comparisons. Considering that beta1 integrin expression is not significantly different between various subtype, would it be possible to conclude that beta1 signaling is independent of TLN1 in the subsets, where TLN1 is not amplified.

2. TNBC samples with high TLN1: are they mesenchymal type? The authors should check the correlation between TLN1/EMT/Hippo pathways, this will strengthen their hypothesis.

3. Figure 2: the conclusion that TLN1 affect stemness, needs to be supported by tumor initiation data. Is there any difference in tumor onset in figure 2E? Also, CSC are slow growing, which is in contrast to the current data, which suggest TLN1 promoted cell proliferation.

4. Figure 2: To validate the role of YAP/TAZ, the authors need to perform rescue experiment using constitutively active YAP/TAZ.

5. Figure 3: Are the effects on EMT proteins mediated by YAP/TAZ patwhays or this is an independent signaling from beta1 integrin.

6. Figure 3: Effect on TNBC cell adhesion and metastasis can be a consequence of reduced cell proliferation; the conclusions needs re-wording.

7. Figure 4: Conclusions on cell adhesion and migration should take viability into considerations. If the cells are dead, they will not adhere or migrate.

8. Figure 4J: western blots are of very poor quality. Bands were cropped too close, should show full blots. Also, inhibitors impacted the expression of beta1 integrin (4I), it will be hard to conclude that reduced association is not due to reduced expression.

9. To further strengthens the conclusion, it is suggested to include cell lines from non-TNBC subtype or normal epithelial in some of their experiments as a control.

*Reviewer #2:*

The article started with bioinformatic and pathological analysis of TCGA and/or breast cancer tissues related to focal adhesion molecules and identified talin-1 (TLN-1) as a worse prognostic marker in breast cancer in particular TNBC. Talin is known to be a component within focal adhesion and binds to the intracellular tail of integrins. Using shRNA-based silencing of TLN-1, the group found that TLN-1 is important in promoting different properties of TNBC cells – including the cell cycle progression, migration/invasion, and cancer-stem cell properties – by reducing integrin-β-mediated intracellular signaling transduction. Most importantly the group found that silencing TLN-1 reduces both primary tumor growth and metastasis, supporting the critical role of TLN-1 in TNBC progression. Using computer-based screening/docking approach, the group further identified the interface blocker compound C67399 that can interfere with the interaction between TLN-1 and integrin and reduce tumor growth and metastasis. The article is well-formulated in general to articulate the critical and new roles of TLN-1 in TNBC and the small inhibitor has potential for further translational studies.

This is in general well-written article that may generate interest in developing TLN-1/integrin inhibitors for TNBC therapy. At the current stages, however, there are some major concerns related to the evidence that can fully support the claims.

1) The justification for TNBC is not supported by the data. Both TNBC and luminal cancer have > 40% positivity and overall DFS showed trend of poor prognosis. The Figure 1D for TNBC DFS should be removed. With such limited number (n=12 low 16 high), I don't think the statistics mean much and only one outlier will damage the statistics.

2) The claims about cancer stem cells are not necessary nor sufficiently supported. Cancer stem cells should be based on in vivo experiments rather than in vitro experiments to look at serial dilutions (cancer forming efficiency) in mouse and calculate tumor forming efficiency. Mammosphere is not accurate assay for cancer stem cell activity, nor CD24 and CD44 double staining. The spheroid structures are not mammospheres (real mammospheres should have compact morphology and nicely arranged ball like structures). Figure 2G looks like more of cell aggregate. The numbers are ranging from 4-7 and there is no description of any related methods anywhere in the article.

3) All the IP experiments are critical to show the interaction between TLN1 and integrin beta1, but most of these western blotting showed very fuzzy bands that could be just non-specific protein blotting, including Figure 4A; Figure 5J.

4) The quality of immune fluorescence in Figure 4 is in very poor quality and it doesn't seem like a co-focal image. There is no visible co-localization based Figure 4B.

5) Figure 4C looks like stress fiber formation after TLN1 is reduced. Rho family GTPases should be checked for activity.

6) shRNA to TLN1 did not show signs of cell death, and yet there is significant reduction in cell viability by the C67399. This raises the question of specificity. C67399 should be used to treat shRNA expressing cells for viability to confirm whether this is on target effect. The same goes to animal studies. This is the biggest issue of this paper to claim the effect of C67399 is through TLN1. The interruption of TLN1 and integrin-b1 is not significant in Figure 6J. There is no explanation why the inhibitor will reduce integrin-b1 expression. There is no biological experiments to show whether C67399 has any effect on TLN1-null or silenced cells.

7) There is no other critical control for the impact of C67399 on different integrins. There are quite several integrin inhibitors available. There are also antibodies that can detect activated integrins. To the least, the paper should address whether C67399 has direct impact on integrin activity and should be compared with other integrin inhibitors for different effect.

*Reviewer #3:*

The authors present here a manuscript detailing the effect of TLN1 gene silencing in breast cancer, and its effects on tumour biology and growth. The authors utilize a combination of in vitro assays and animal models of breast cancer (although limited in number). Further, they expand their efforts using an in-silico analysis of a large drug library to identify a lead compound that can impair TLN1 gene function (with integrin β 1) and provide preliminary evidence of its utility to affect TNBC cell and tumour growth.

The data are well presented, however the manuscript (in its current form) requires more detail to support their hypotheses and conclusions; specifically, in terms of expanded methodology, grammatical review, reanalysis of tumour growth rate, and an increased volume of cell line and patient data that can better delineate whether TLN1 is truly overrepresented in TNBC (compared to other molecular subtypes, or TNBC subtypes), and whether TLN1 biology is similar amongst these subtypes (which would increase the relevance of this paper), or if it is selective to TNBC (which would still be of importance as this is aggressive class of breast cancer with limited therapeutic options).

To bolster the strength of this manuscript, please consider the following:

Figure 1A Only 3 of 26 known integrin genes were involved in the analysis. This makes the current finding vague and low powered. Consider expanding the analysis to all 26 integrin genes, since the data should be available in publicly accessible gene expression data sets. Would be of interest to correlate gene expression with, not only molecular subclasses, but also survival metrics including relapse-free, distant metastasis and overall survival. Further, combining TLN1+/- expression with various segregation of integrin expression may yield supportive data.

Figure 1C Only 4 TNBC samples used for validation; no other breast cancer subtypes. Expand to be more inclusive of ER+ and HER2+ tumours.

Figure 1D Please show DFS in all molecular subclasses (ER+ and HER2+) of breast cancer. Provide insight into whether this is a TNBC-only phenomenon, or potentially similarly effective in all subclasses. Without this, it is not clear why the Authors would be selective only for TNBC.

Sup Figure 1A. Does TLN2 mRNA similarly stratify DFS in all breast cancer molecular subtypes, as TLN1 does? Again, without this data it is not clear why the Authors would be selective only for TLN1, and also TNBC.

Figure 2 Should incorporate a more extensive panel of breast cancer cell lines. Only 4 are shown which severely limits the correlation that TLN1 is overrepresented in TNBC. Expand panel to include at least 10 TNBC and minimum 5 ER+ and 5 HER2+ cell lines. Potentially also include mRNA gene expression within cell lines and their molecular subtypes (see Neve et al., 2006; accessible through http://co.bmc.lu.se/gobo/); which does seem to suggest TLN1 overexpression in TNBC in a panel of 50+ cell lines. A further point would be to stratify TLN1 expression across subtypes of TNBC according to Lehman et al., 2011; (https://www.jci.org/articles/view/45014). IS TLN1 predominantly expressed in TNBC subtypes that are more mesenchymal (and that inherently have a more "stem-like" phenotype). A better understanding of the background of where TLN1 is overexpressed will improve this manuscript, in terms of its relevance and importance.

Figure 2E a statistical evaluation of tumour growth rates should be performed using regression analysis, and not measured based on tumour volume size between groups at each time point. It is important to establish that the loss of TLN1 leads to a reduction in tumour growth rate, and not just tumour size at independent time points (as this could be affected by differences in tumour inoculation). What about spontaneous lung metastasis?

Figure 2. What are the effects of TLN1 silencing in other TLN1+ TNBC cell lines (or ER+ cell lines?) Are similar effects observed in cell-cycle, growth, and stem-like properties. As 231 cells are largely all "stem like", it would be important to assess this in other cell lines where "stem-like" phenotype is less extensive.

Figure 2. Does restoration of Hippo/Yap signalling pathway lead functionally restore TLN1-mediated growth inhibition in TNBC cell lines?

Figure 3 Effect on lung metastasis could just be due to reduced growth rate of modified 231 cell lines. Very little information is given in this manuscript about the experimental tail vein experiment shown in Figure 3E. How many cells were inoculated into the tail vein? How long did the experiment go for? Was routine bioluminescent imaging performed so that a true evaluation of metastatic growth rate could be determined? Do lung metastases form in animals inoculated with 231-shRNA-TLN1? If they didn't, can the experiment be extended until they do? A Kaplan-Meier analysis of survival of time to metastasis combined with bioluminescent imaging is the gold standard for experimental lung metastasis assays. Without it, no inference on metastasis can be made (and is also there is no data on metastasis shown from the orthotopically-implanted animal experiment in Figure 2). Please clarify.

Figure 4 Does TLN1 biology (interactions with beta1 integrin) similarly occur in ER+ breast cancer, or other TNBC cell lines? Does it lead to similar effects on cell localization and changes in phosphor AKT, FAK, etc? Or is it a specific TNBC effect?

Figure 5 Does C67399 similarly affect cell stemness by flowcytometry with CD24/CD44 or ALDH1+? Does it affect mammosphere formation? Does C67399 similarly affect YAP/TAZ/Oct4 signalling? Or is it just killing cancer cells, with no link to stemlike actions of TLN1? What are the effects of C67399 in vitro in TLN1 negative breast cancer cell lines?

Figure 6 Please separate out the results of the two animal experiments; it is unclear whether the lung metastasis is spontaneous (from the mammary gland injected cancer cells), or from the experimental metastasis tail vein assay. Again, the details for the experimental metastasis tail vein assay are lacking, please see my comments above. Since TLN1 is associated with chemosensitivity, would a combination of C67399 with standard of care chemotherapy drugs be more effective in the in vivo studies? Is C67399 similarly effective on TNBC xenografts (and/or other molecular subtypes of breast cancer) that do not express TLN1?

---

## [Author Response]

Reviewer #1:The authors established TLN1 as a potential therapeutic target for triple-negative breast cancer, an aggressive subtype, with very few treatment options. The authors used IHC on patient samples and concluded that high TLN1, but not TLN2 and other integrins expression is an independent prognostic marker for BrCa. It is important to note that TLN2/integrin expression was studied in TCGA dataset and not on the same set of samples and the conclusions on TLN2/integrins should be re-considered. The authors presented strong data on effect of TLN downregulation on cell proliferation, migration, invasion, stemness, EMT and in vivo tumor growth and metastasis. Their data suggest all aspects of tumor progression are impacted by TLN1 downregulation. Mechanistically, TLN1 downregulation impacted LATS/YAP/TAZ pathway as well as EMT proteins. The authors concluded that TLN1 silencing reduced tumor growth through Hippo/Yap signaling, which is not justified as there was no data presented, where activation of YAP/TAZ rescued stemness/tumor growth in TLN1 ablated cells. Presented data is just correlational. Same is true regarding their conclusion on EMT and Metastasis. Next, the authors confirmed the effect of TLN1 alteration on focal adhesions formation as well as other integrin signaling pathways. In the last part of the manuscript, the authors described the development and characterization of small molecule, which can inhibit interaction between beta1 integrin and TLN1. This inhibitor, C67399, affected viability, adhesion, migration, tumor growth and metastasis. However, it is important to point that all these effect on malignancy can be explained by the viability. Since cells were less viable, it is likely that non-viable cells will not adhere, migrate, form tumor or mets. This is an interesting article, but text needs to be re-worded to justify conclusions based on the data presented.

Many thanks for this critical and helpful comment! TLN1 is expressed in almost all tissues, while TLN2 is usually expressed mainly in the heart, brain, testis and muscle (23266827; 19220457). TLNs are located in a complex between adherent cells and their extracellular matrix (ECM) and regulate integrative and adhesive signaling. Previous analysis revealed that TLN1 wast the most highly expressed integrin cytoskeleton cross-linker that can trigger integrin activation (21547905). TLN1 overexpression promotes tumor invasion and metastasis via focal adhesion (20160039). A key event is binding to the integrin β cytoplasmic tail by TLN (14526080; 17627302; 18086863), a 270 kDa protein (capable of forming homodimers) with an N-terminal head domain (comprising F0, F1, F2, and F3 subdomains) and a C-terminal rod domain that binds to vinculin and actin (18434644). Binding of the F3 domain to integrin β tails is sufficient for integrin activation (11932255), although other head domains contribute to activation (18165225). Furthermore, TLN1 phosphorylation activates β1 integrins to promote prostate cancer bone metastasis (24793790).

Recommendations for the authors:1. Figure 1: To invalidate the role of TLN2/integrins, the authors should either use same patient samples with IHC for TLN2 or use TCGA for TLN1 comparisons. Considering that beta1 integrin expression is not significantly different between various subtype, would it be possible to conclude that beta1 signaling is independent of TLN1 in the subsets, where TLN1 is not amplified.

We appreciate for this very critical comment from the referee. We would like to comment on this from two folds. (1). We first compare the AA sequence of F3 domain ring-region between TLN1 and TLN2. Please see the superimposition Author response image 1, (TLN1: aa 324-370; TLN2: aa 327-373).

**Author response image 1. sa2fig1:** 

Moreover, please see a more detailed superimposition between TLN1 and TLN2 Author response image 2 (Sci Rep. 2017;7:41989),

Therefore, we conclude that there is a rather significant difference between F3 domain ring-region of TLN1 and TLN2, where beta1 integrin interacts with TLN subtypes. We agree with the referee that it is possible that beta1 signaling is independent of TLN1 in the subsets, where TLN1 is not amplified. (2). Next, we compute and examine the binding affinities of C67399 small-molecule compound with TLN1 and TLN2, respectively. Our calculation results implicate that c67399 small-molecule compound binds more strongly with TLN1 F3 domain as compared to the binding with TLN2 F3 domain. Noteworthy, there is a nearly 1kal/mol energy difference (corresponding to more than 10-fold in terms of binding constant) and therefore c67399 small-molecule compound selectively binds with TLN1 F3 domain. Moreover, our results suggest that c67399 small-molecule might be a selective inhibitor which interrupts the interaction of TLN1 F3 domain with beta1 integrin.

**Author response image 3. sa2fig3:** 

Furthermore, although the expression of integrin β1 is not significantly different between various subtype, the correlation between TLN1 and integrin β1 by GEPIA showed a significant positive correlation (p<0.01, R=0.34). Therefore, with the present results, we cannot conclude that integrin β1 signaling is independent of TLN1 in the isoforms in which TLN1 is not amplified.

**Author response image 4. sa2fig4:** 

2. TNBC samples with high TLN1: are they mesenchymal type? The authors should check the correlation between TLN1/EMT/Hippo pathways, this will strengthen their hypothesis.

We appreciate for the comment from the referee. We are not sure if all TNBC samples used in IHC are of the mesenchymal types. However, it has been shown that the JAK/STAT3 signaling pathway is upregulated in TNBC mesenchymal types, that this subtype has higher expression of JAK1 and IL6, which are important drivers of JAK/STAT3 activation, and that activation or tyrosine phosphorylation of STAT3 (pSTAT3) gene signature scores (26234940) are higher in mesenchymal types than in other subtypes. Therefore, we analyzed the correlation of TLN1 with key regulators in the JAK/STAT3 pathway by GEPIA (gene expression profiling interactive analysis, http://gepia.cancer-pku.cn/) and found that TLN1 was positively correlated with JAK1, IL6 and STAT3 were positively correlated (p<0.01 for all, R=0.23 for JAK1 and STAT3, and R=0.1 for IL6), suggesting that TLN1 high expression is largely a TNBC mesenchymal type.

**Author response image 5. sa2fig5:** 

In addition, we also analyzed the correlation between TLN1 and key regulators in the EMT/Hippo pathway with the help of GEPIA, and found that TLN1 was negatively correlated with CDH1 (p<0.01, R=-0.11), positively correlated with CDH2 (p=0.12, R=0.048), VIM (p<0.01, R=0.51), VCL(p<0.01, R=0.31) and SNAI1 (p<0.01, R=0.27). Moreover, TLN1 was positive correlation with YAP1 (p<0.01, R=0.28) and TAZ (p<0.01, R=0.17), thus indicating the positive correlation of TLN1 with EMT/Hippo pathway. Per the referees' comments, we decided to remove the data on CSC and YAP/TAZ pathway from this manuscript to avoid ambiguity.

**Author response image 6. sa2fig6:** 

3. Figure 2: the conclusion that TLN1 affect stemness, needs to be supported by tumor initiation data. Is there any difference in tumor onset in figure 2E? Also, CSC are slow growing, which is in contrast to the current data, which suggest TLN1 promoted cell proliferation.

We want to thank the reviewer for this suggestion. There was no significant difference in tumor incidence between 231/NC and 231/shTLN1 in Figure 2E, but the overall decrease in tumor size in the 231/shTLN1 group suggests that TLN1 knockdown affects tumor growth. In addition, stem cells grow rapidly in specific stem cell medium, which is enriched with cytokines that maintain stem cell growth, and our current data show a significant downregulation of the proportion of CD44+CD24- CSCs after knockdown of TLN1 and a slowed proliferation and reduced sphere-forming ability in stem cell medium. Per the referees' comments, we decided to remove the data on CSC from this manuscript to avoid ambiguity.

4. Figure 2: To validate the role of YAP/TAZ, the authors need to perform rescue experiment using constitutively active YAP/TAZ.

We are thankful for this critical comment. Per the referees' comments, we decided to remove the data on CSC and YAP/TAZ from this manuscript to avoid ambiguity.

5. Figure 3: Are the effects on EMT proteins mediated by YAP/TAZ patwhays or this is an independent signaling from beta1 integrin.

We are thankful for this critical comment. The effects of EMT proteins have been reported in the literature to be regulated both by YAP1/TAZ patwhays (31927328, 33421513) and by integrin β1 (27836001). YAP1/TAZ proteins can then receive regulatory input from tight junctions and polar complexes, as well as proteins from adherens junctions (22075429). In this study, we focus on the fact that TLN1 binding to integrin β1 activates both focal adhesion to regulate cell-ECM adhesion and integrin downstream signaling pathways, as well as cell-cell contacts, ultimately leading to cell migration and tumor metastasis in concert. Per the referees' comments, we decided to remove the data on CSC and YAP/TAZ pathway from this manuscript to avoid ambiguity.

6. Figure 3: Effect on TNBC cell adhesion and metastasis can be a consequence of reduced cell proliferation; the conclusions needs re-wording.

We want to thank the reviewer for this very insightful comment. To exclude the possibility that the metastasis reduction following knockdown of TLN1 on TNBC cells was due to reduced cell proliferation, we performed tail vein injection of MDA-231 cells to form experimental lung metastases in Balb/c-nu mice (M Shen, 2021) and found a significant reduction in lung metastasis after TLN1 knockdown.

7. Figure 4: Conclusions on cell adhesion and migration should take viability into considerations. If the cells are dead, they will not adhere or migrate.

Many thanks for the critical comments. Based on the recommendations of the referee, we describe the above findings more critically, and the effect on TNBC cell adhesion and metastasis may also be influenced by reduced cell proliferation. To exclude this effect, we performed an in vivo test in a tail vein injection lung metastasis model, and the results showed that knockdown of TLN1 inhibited lung metastasis.

8. Figure 4J: western blots are of very poor quality. Bands were cropped too close, should show full blots. Also, inhibitors impacted the expression of beta1 integrin (4I), it will be hard to conclude that reduced association is not due to reduced expression.

We want to thank the reviewer for this comment. According to the referee’s recommendation, we add the display of the IP complete blots. In addition, combining the westernbot results and IP results of C67399 on integrin β1, it can be seen that C67399 affects both the partial expression of integrin β1 and the binding of TLN1 to integrin β1 (Figure 4-source data 1 and Figure 5-source data 1).

9. To further strengthens the conclusion, it is suggested to include cell lines from non-TNBC subtype or normal epithelial in some of their experiments as a control.

Thanks to the referee for the critical comments. From Figure 2A, we can see that TLN1 is low expressed in MCF7 and SK-BR-3 cells, thus we focus on the effect of TLN1 in TNBC cells rather than other types of breast cancer cell line.

We are truly grateful for your kind help and guidance. Again, we sincerely appreciate the time and effort you have spent on reviewing our manuscript.

Reviewer #2:The article started with bioinformatic and pathological analysis of TCGA and/or breast cancer tissues related to focal adhesion molecules and identified talin-1 (TLN-1) as a worse prognostic marker in breast cancer in particular TNBC. Talin is known to be a component within focal adhesion and binds to the intracellular tail of integrins. Using shRNA-based silencing of TLN-1, the group found that TLN-1 is important in promoting different properties of TNBC cells – including the cell cycle progression, migration/invasion, and cancer-stem cell properties – by reducing integrin-β-mediated intracellular signaling transduction. Most importantly the group found that silencing TLN-1 reduces both primary tumor growth and metastasis, supporting the critical role of TLN-1 in TNBC progression. Using computer-based screening/docking approach, the group further identified the interface blocker compound C67399 that can interfere with the interaction between TLN-1 and integrin and reduce tumor growth and metastasis. The article is well-formulated in general to articulate the critical and new roles of TLN-1 in TNBC and the small inhibitor has potential for further translational studies.This is in general well-written article that may generate interest in developing TLN-1/integrin inhibitors for TNBC therapy. At the current stages, however, there are some major concerns related to the evidence that can fully support the claims.Major concerns:1) The justification for TNBC is not supported by the data. Both TNBC and luminal cancer have > 40% positivity and overall DFS showed trend of poor prognosis. The Figure 1D for TNBC DFS should be removed. With such limited number (n=12 low 16 high), I don't think the statistics mean much and only one outlier will damage the statistics.

Thanks to the referee for the critical comments. Triple-negative breast cancer lacks effective targets, while hormone receptor-positive breast cancer has anti-endocrine therapeutic targets, so we prioritize the study of triple-negative breast cancer. To strengthen the credibility of the data, we expanded the cases of triple-negative breast cancer to 171 cases and statistically found that TLN1 in TNBC still showed a poor survival prognosis (Figure 1D).

2) The claims about cancer stem cells are not necessary nor sufficiently supported. Cancer stem cells should be based on in vivo experiments rather than in vitro experiments to look at serial dilutions (cancer forming efficiency) in mouse and calculate tumor forming efficiency. Mammosphere is not accurate assay for cancer stem cell activity, nor CD24 and CD44 double staining. The spheroid structures are not mammospheres (real mammospheres should have compact morphology and nicely arranged ball like structures). Figure 2G looks like more of cell aggregate. The numbers are ranging from 4-7 and there is no description of any related methods anywhere in the article.

Thanks to the referee for the critical comments. After taking into account the referee' s suggestions, we decided to remove the cancer stem cell content, including mammospheres and double staining for CD24 and CD44.

3) All the IP experiments are critical to show the interaction between TLN1 and integrin beta1, but most of these western blotting showed very fuzzy bands that could be just non-specific protein blotting, including Figure 4A; Figure 5J.

Thanks to the referee for the critical comments. Since IP is essential to show the interaction between TLN1 and integrin β1, we supplementally show the full membrane results of all IP experiments, including Figure 4A and Figure 5J (Figure 4-source data 1 and Figure 5-source data 1).

4) The quality of immune fluorescence in Figure 4 is in very poor quality and it doesn't seem like a co-focal image. There is no visible co-localization based Figure 4B.

Thanks to the referee for the critical comments. Per the referee’s comment, we re-conduncted the staining and showed new confocal results in Figure 4B.

5) Figure 4C looks like stress fiber formation after TLN1 is reduced. Rho family GTPases should be checked for activity.

Thanks to the referee for the critical comments. The reduction of stress fibers after TLN1 knockdown is consistent with previous literature descriptions (32273388, 27043085), but Rho family GTPases and stress fibers are not the argument of interest in this study, we are more concerned with the effect on focal adhesion after TLN1 knockdown.

6) shRNA to TLN1 did not show signs of cell death, and yet there is significant reduction in cell viability by the C67399. This raises the question of specificity. C67399 should be used to treat shRNA expressing cells for viability to confirm whether this is on target effect. The same goes to animal studies. This is the biggest issue of this paper to claim the effect of C67399 is through TLN1. The interruption of TLN1 and integrin-b1 is not significant in Figure 6J. There is no explanation why the inhibitor will reduce integrin-b1 expression. There is no biological experiments to show whether C67399 has any effect on TLN1-null or silenced cells.

Thanks to the referee for the critical comments. Figure 2C-F and shows that shTLN1 affects cell proliferation ability, apoptosis and tumor grouth, while C67399 causes a significant decrease in cell viability, indicating that C67399 is able to inhibit TLN1 expression exerting an inhibitory effect on cell proliferation. In Fig5J, C67399 was able to inhibit the expression of TLN1, while TLN1 was able to affect the expression of intergrin β1, so C67399 inhibited the expression of intergrin β1. For TNBC cells that do not express TLN1, C67399 did not have a significant inhibitory effect.

7) There is no other critical control for the impact of C67399 on different integrins. There are quite several integrin inhibitors available. There are also antibodies that can detect activated integrins. To the least, the paper should address whether C67399 has direct impact on integrin activity and should be compared with other integrin inhibitors for different effect.

We want to thank the referee for the critical comments. As added in the Discussion section, C67399, an inhibitor against TLN1-integrin β1 in this study, showed inhibition of TLN1 and thus tumor growth and metastasis, and it has been reported in the literature that TLN1 phosphorylation activates integrin β1 to promote bone metastasis in prostate cancer (24793790). However, the inhibition of TLN1 phosphorylation and activation of integrin b1 by C67399 is not known.

Reviewer #3:The authors present here a manuscript detailing the effect of TLN1 gene silencing in breast cancer, and its effects on tumour biology and growth. The authors utilize a combination of in vitro assays and animal models of breast cancer (although limited in number). Further, they expand their efforts using an in-silico analysis of a large drug library to identify a lead compound that can impair TLN1 gene function (with integrin β 1) and provide preliminary evidence of its utility to affect TNBC cell and tumour growth.The data are well presented, however the manuscript (in its current form) requires more detail to support their hypotheses and conclusions; specifically, in terms of expanded methodology, grammatical review, reanalysis of tumour growth rate, and an increased volume of cell line and patient data that can better delineate whether TLN1 is truly overrepresented in TNBC (compared to other molecular subtypes, or TNBC subtypes), and whether TLN1 biology is similar amongst these subtypes (which would increase the relevance of this paper), or if it is selective to TNBC (which would still be of importance as this is aggressive class of breast cancer with limited therapeutic options).To bolster the strength of this manuscript, please consider the following:Figure 1A Only 3 of 26 known integrin genes were involved in the analysis. This makes the current finding vague and low powered. Consider expanding the analysis to all 26 integrin genes, since the data should be available in publicly accessible gene expression data sets. Would be of interest to correlate gene expression with, not only molecular subclasses, but also survival metrics including relapse-free, distant metastasis and overall survival. Further, combining TLN1+/- expression with various segregation of integrin expression may yield supportive data.

We want to thank the reviewer for the helpful and critical comments. Per the comments, we have performed and expanding analysis to all integrins. We hope these could address the concerns.

**Author response image 7. sa2fig7:** TLN1 with three integrins in the manuscript.

**Author response image 8. sa2fig8:** TLN1 with 31 integrins.

**Author response image 9. sa2fig9:** TLN1& ITGB1.

**Author response image 10. sa2fig10:** 

**Author response image 11. sa2fig11:** 

Figure 1C Only 4 TNBC samples used for validation; no other breast cancer subtypes. Expand to be more inclusive of ER+ and HER2+ tumours.

Thanks to the referee for the critical comments. As mentioned previously, this study found that TLN1 protein was specifically highly expressed in TNBC cells compared to other molecular types, suggesting a unique potential role of TLN1 for TNBC and therefore focusing on TNBC.

Figure 1D Please show DFS in all molecular subclasses (ER+ and HER2+) of breast cancer. Provide insight into whether this is a TNBC-only phenomenon, or potentially similarly effective in all subclasses. Without this, it is not clear why the Authors would be selective only for TNBC.

Thanks to the referee for the critical comments. As mentioned previously, this study found that TLN1 protein was specifically highly expressed in TNBC cells compared to other molecular subtypes, suggesting a unique potential role of TLN1 for TNBC, while limiting the expression of TLN2 in tissues, so this study focused on the intervention effect of TLN1 with TNBC and its compound C67399. To strengthen the credibility of the data, we expanded the cases of triple-negative breast cancer to 171 cases and statistically found that TLN1 in TNBC still showed a poor survival prognosis (Figure 1D).

Sup Figure 1A. Does TLN2 mRNA similarly stratify DFS in all breast cancer molecular subtypes, as TLN1 does? Again, without this data it is not clear why the Authors would be selective only for TLN1, and also TNBC.Figure 2 Should incorporate a more extensive panel of breast cancer cell lines. Only 4 are shown which severely limits the correlation that TLN1 is overrepresented in TNBC. Expand panel to include at least 10 TNBC and minimum 5 ER+ and 5 HER2+ cell lines. Potentially also include mRNA gene expression within cell lines and their molecular subtypes (see Neve et al., 2006; accessible through http://co.bmc.lu.se/gobo/); which does seem to suggest TLN1 overexpression in TNBC in a panel of 50+ cell lines. A further point would be to stratify TLN1 expression across subtypes of TNBC according to Lehman et al., 2011; (https://www.jci.org/articles/view/45014). IS TLN1 predominantly expressed in TNBC subtypes that are more mesenchymal (and that inherently have a more "stem-like" phenotype). A better understanding of the background of where TLN1 is overexpressed will improve this manuscript, in terms of its relevance and importance.

Per the referee’s suggestion, we have incorporate a more extensive panel of breast cancer cell lines into the analysis as Author response image 12. We hope this could address the concerns.

**Author response image 12. sa2fig12:** 

Figure 2E a statistical evaluation of tumour growth rates should be performed using regression analysis, and not measured based on tumour volume size between groups at each time point. It is important to establish that the loss of TLN1 leads to a reduction in tumour growth rate, and not just tumour size at independent time points (as this could be affected by differences in tumour inoculation). What about spontaneous lung metastasis?

We are thankful for this critical comment. No spontaneous pulmonary metastases were detected due to the rapid growth of the primary focus.

Figure 2. What are the effects of TLN1 silencing in other TLN1+ TNBC cell lines (or ER+ cell lines?) Are similar effects observed in cell-cycle, growth, and stem-like properties. As 231 cells are largely all "stem like", it would be important to assess this in other cell lines where "stem-like" phenotype is less extensive.

Thanks to the referee for the critical comments. In this study, TLN1 protein was found to be specifically highly expressed in TNBC cells compared to other molecular subtypes, suggesting a unique potential role of TLN1 for TNBC and therefore focusing on TNBC.

Figure 2. Does restoration of Hippo/Yap signalling pathway lead functionally restore TLN1-mediated growth inhibition in TNBC cell lines?

Thanks to the referee for the critical comments. Per the referees' comments, we decided to remove the data on CSC and YAP/TAZ pathway from this manuscript to avoid ambiguity.

Figure 3 Effect on lung metastasis could just be due to reduced growth rate of modified 231 cell lines. Very little information is given in this manuscript about the experimental tail vein experiment shown in Figure 3E. How many cells were inoculated into the tail vein? How long did the experiment go for? Was routine bioluminescent imaging performed so that a true evaluation of metastatic growth rate could be determined? Do lung metastases form in animals inoculated with 231-shRNA-TLN1? If they didn't, can the experiment be extended until they do? A Kaplan-Meier analysis of survival of time to metastasis combined with bioluminescent imaging is the gold standard for experimental lung metastasis assays. Without it, no inference on metastasis can be made (and is also there is no data on metastasis shown from the orthotopically-implanted animal experiment in Figure 2). Please clarify.

Thanks to the referee for the critical comments. To rule out that the effect on lung metastasis was only due to the reduced growth rate of the modified 231 cell line, in this manuscript we used tail vein alone experiments to see if knockdown of TLN1 affects lung metastasis, with specific methods added to the methodology. Bioluminescence imaging was not performed in this study, so that lung metastasis could only be determined after final execution of the sampling. Due to the rapid growth of submammary fat pad inoculated MDA-MB-231 cells, it was too late to observe spontaneous lung metastasis.

Figure 4 Does TLN1 biology (interactions with beta1 integrin) similarly occur in ER+ breast cancer, or other TNBC cell lines? Does it lead to similar effects on cell localization and changes in phosphor AKT, FAK, etc? Or is it a specific TNBC effect?

We are thankful for this critical comment. In this study, TLN1 protein was found to be specifically highly expressed in TNBC cells compared to other molecular subtypes, suggesting a unique potential role of TLN1 for TNBC and therefore focusing on TNBC, rather than other molecular subtypes.

Figure 5 Does C67399 similarly affect cell stemness by flowcytometry with CD24/CD44 or ALDH1+? Does it affect mammosphere formation? Does C67399 similarly affect YAP/TAZ/Oct4 signalling? Or is it just killing cancer cells, with no link to stemlike actions of TLN1? What are the effects of C67399 in vitro in TLN1 negative breast cancer cell lines?

We are thankful for this critical comment. Per the referees' comments, we decided to remove the data on CSC and YAP/TAZ pathway from this manuscript to avoid ambiguity.

Figure 6 Please separate out the results of the two animal experiments; it is unclear whether the lung metastasis is spontaneous (from the mammary gland injected cancer cells), or from the experimental metastasis tail vein assay. Again, the details for the experimental metastasis tail vein assay are lacking, please see my comments above. Since TLN1 is associated with chemosensitivity, would a combination of C67399 with standard of care chemotherapy drugs be more effective in the in vivo studies? Is C67399 similarly effective on TNBC xenografts (and/or other molecular subtypes of breast cancer) that do not express TLN1?

We want to thank the reviewer for this very insightful comment. We have added detailed information on experimental tail vein injection experiments in the method blood. In addition, this study focused on ths development of novel targeted agents for the TLN1-integrin β1 binding site and validation of their inhibitory effects on tumor growth and metastasis, and did not address chemosensitivity. For TNBC cells that do not express TLN1, C67399 did not have a significant inhibitory effect.

We are grateful for your insightful comments and guidance. Again, we sincerely appreciate the time and effort you have spent on reviewing our manuscript.